# Constraining net long term climate feedback from satellite observed internal variability possible by mid 2030s

Alejandro Uribe[1,2], Frida A.-M Bender[1,2], and Thorsten Mauritsen[1,2]

[1]Department of Meteorology, Stockholm University, Stockholm, Sweden
[2]Bolin Centre for Climate Research, Stockholm, Sweden

**Correspondence:** Alejandro Uribe (alejandro.uribe-cortes@misu.su.se)

**Abstract.** Observing climate feedbacks to long-term global warming, which are crucial climate regulators, isn´t feasible within the observational record. However, linking them to top-of-the-atmosphere flux variations in response to natural surface temperature fluctuations (internal variability feedbacks) is a viable approach. We explore the use of relating internal variability to forced climate feedbacks in models and applying the resulting relationship to observations to constrain forced climate feed-
backs. Our findings reveal strong longwave and shortwave feedback relationships in models during the 14-year overlap with the CERES record. Yet, due to the weaker relationship between internal variability and forced climate longwave feedbacks, the net feedback relationship remains weak, even over longer periods beyond the CERES record. However, after about half a century, this relationship strengthens primarily due to reinforcements of the internal variability and forced climate short-wave feedback relationship. We therefore explore merging the satellite records with reanalysis to establish an extended data
record. The resulting constraint suggests a stronger negative forced climate net feedback than the model´s distribution and an equilibrium climate sensitivity of about 2.59 K (1.95 K to 3.12 K, 5-95% confidence intervals). Nevertheless, this method doesn't account for certain factors like biogeophysical-chemical feedbacks, inactive on short time-scales and not represented in most models, along with differences in historical warming patterns, which may lead to misrepresenting climate sensitivity. Additionally, continuous satellite observations until at least the mid-2030s are essential for using purely observed estimates of
the net internal variability feedback to constrain the net forced climate feedback and, consequently, climate sensitivity.

## 1   Introduction

The increasing levels of atmospheric carbon dioxide have significant implications for the Earth's climate system. Elevated carbon dioxide concentrations enhance the absorption and emission of infrared radiation in the atmosphere, leading to a radiative imbalance at the Top of the Atmosphere (TOA) and subsequent warming of the troposphere (NRC, 1979).
The long-term response of global temperatures to this radiative forcing is influenced by a complex interplay of mechanisms, where the initial radiative forcing initiates changes in secondary processes that, in turn, impact the original forcing. These mechanisms, known as forced climate feedbacks, involve processes such as temperature, water vapor, lapse rate, surface albedo, clouds, biogeophysical, and biochemical (Forster et al., 2021). These feedback mechanisms play a crucial role in either amplifying or attenuating the initial warming signal and collectively determine the Equilibrium Climate Sensitivity

(ECS), which quantifies the global temperature response to a doubling of atmospheric carbon dioxide concentrations relative to pre-industrial levels.

Generally, in estimating forced climate feedbacks, tools such as theory, observations, climate models, and fine-scale simulations are commonly utilized (Sherwood et al., 2020). Climate models are particularly important as they are designed to solve the complex equations governing the Earth's climate system. However, climate models need to parameterize unresolved processes by establishing empirical relationships with explicitly resolved variables (Williamson et al., 2021). While climate models share fundamental equations, the use of different parameterization approaches among them introduces variations in future projections, including forced climate feedbacks, leading to a range of model-estimated ECS. Assessing how models represent forced climate feedbacks is challenging due to the absence of long-term global observations spanning decades or centuries. Consequently, reducing uncertainty in model-estimated forced climate feedbacks becomes a complex task. However, if a significant and physically explainable relationship emerges across climate models between observable properties of the climate system and forced climate change, past climate observations can be employed to constrain these feedbacks.

Several studies have identified relationships between various observables and forced climate feedbacks across different generations of models (Forster et al., 2021). For instance, elements of the present mean state have been utilized (e.g., Trenberth and Fasullo, 2010; Brient et al., 2016), as well as past climate change (e.g., Hargreaves et al., 2012; Jiménez-de-la Cuesta and Mauritsen, 2019; Renoult et al., 2020) and internal variability (Dessler, 2013; Mauritsen and Stevens, 2015; Dessler et al., 2018). Focusing specifically on climate feedbacks, Loeb et al. (2018) identified a robust relationship between cloudy-sky flux on timescales of 2.5 to 3 years and ECS. However, they noted that 100 years of data are necessary for this relationship to become statistically robust. Using models from the Coupled Model Intercomparison Project phase 6 (CMIP6), Lutsko et al. (2021) found significant relationships between 50 years of cloud variability and regional forced climate cloud feedbacks across most regions, with the exception of a latitudinal band from 60°N to 90°N. Dessler and Forster (2018) used models from the Precipitation Driver Response Model Intercomparison Project (Myhre et al., 2017) to find a relationship between TOA flux changes in response to natural variations of surface temperature (referred here as internal variability feedbacks) in simulations where pre-industrial carbon dioxide levels are maintained over a century and those resulting from the doubling of carbon dioxide concentrations. By incorporating observed internal variability feedbacks spanning from 2000 to 2017 alongside this relationship, they derived an ECS likely range of 2.4–4.6 K (17–83% confidence interval). However, uncertainty persists regarding the appropriateness of integrating observations and the modeled relationship from different time periods to constrain forced climate feedbacks and ECS. Similarly, Uribe et al. (2022) employed CMIP6 models and demonstrated that the strength of forced climate feedbacks is associated with internal variability feedbacks from 2001 to 2014. Nevertheless, they found that this relationship did not hold for net feedback. Additionally, they concluded that uncertainty in simulated and observed internal variability feedbacks over this short period precludes the establishment of an emergent constraint on forced climate feedbacks

In this study, our primary objective is to contribute to the reduction of uncertainties in forced climate feedbacks by advancing our understanding of the use of observations and the relationships between internal variability feedbacks and forced climate feedbacks in CMIP6 models. Building upon previous research (Uribe et al., 2022), we specifically investigate the underlying

factors contributing to the absence of a robust relationship between internal variability and forced climate net feedbacks, despite the evident strong relationships observed for longwave and shortwave feedback components in CMIP6 models. Here, we focus on the historical simulation period that aligns with the available Clouds and the Earth's Radiant Energy System (CERES) satellite data record. In addition, we investigate whether the challenges observed in establishing a relationship between internal variability and forced net climate feedback in models persist when longer historical periods are considered. Subsequently, we explore the suitability of employing distinct timeframes for estimating both model and observed internal variability feedbacks when utilizing observations to constrain net forced feedbacks in models. This examination aims to ascertain the minimum required record length. Finally, we merge satellite observations with reanalysis data to obtain an emergent constraint on forced climate net feedback.

## 2  Materials and Methods

We study the relationship between feedbacks arising from internal variability and external forcing in models participating in CMIP6, specifically in coupled ocean-atmosphere experiments (CMIP), simulating interactions between the ocean and atmosphere, and in atmosphere-only experiments (AMIP), focusing exclusively on atmospheric processes while using prescribed observed sea surface temperatures. To accomplish this, we utilize historical simulations and 150-year experiments where atmospheric carbon dioxide concentrations are abruptly quadrupled from pre-industrial levels and subsequently held constant (abrupt4xCO2). To capture a broader range of possible historical climate outcomes and obtain robust estimates of internal variability feedbacks, we utilize up to 5 realizations of historical ensemble members, incorporating a more extensive set of models compared to the approach used in Uribe et al. (2022) (Table 1).

In order to quantify feedbacks, we use the planetary energy balance at TOA:

$$R = F + \lambda T \tag{1}$$

where $R$ is the net TOA radiative flux anomaly, $F$ is the radiative forcing, $\lambda$ is the radiative feedback parameter and $T$ is the surface temperature anomaly (Gregory et al., 2004; Dessler et al., 2018).

We calculate forced climate feedbacks using linear Ordinary Least Squares (OLS) regression coefficients derived from 150 years of annual global averages of $R$ and $T$ from abrupt4xCO2 simulations. It is important to note that this approach might lead to an underestimation of ECS compared to estimates derived from millennial-length simulations, due to the influence of time-dependent feedbacks (Rugenstein et al., 2020). Conversely, ECS estimates from 150-year abrupt4xCO2 experiments often overestimate ECS compared to those from 2xCO2 experiments, owing to nonlogarithmic forcing, feedback $CO_2$ dependence, and feedback temperature dependence (Bloch-Johnson et al., 2021). Given that examining the interaction between these effects is beyond the scope of our study, we adopted the standard method as it offers a straightforward approach and provides a basis for comparison and analysis.

Similarly, to estimate internal variability feedbacks, we use OLS regression coefficients from linearly detrended (we assume a linear forcing influence in the data series, given the relatively short periods we are analyzing) and deseasonalized monthly global averages anomalies of $R$ and $T$. However, given that monthly temperature time series can display temporal autocorrelation, which may affect the estimation of regression coefficients and standard errors, we took specific steps to address this issue. For observational temperature data, we calculated the autocorrelation function and adjusted the degrees of freedom for standard error computations accordingly. For model data, we employed an extended OLS approach (Generalized Least Squares). Specifically, we transformed each temperature time series per model realization into a set of weighted variables. The weights were determined by fitting autoregression models of order one to account for autocorrelation. We then combined all transformed temperature time series realizations for each model and applied the OLS regression to estimate the regression coefficients, yielding a single estimate of internal variability feedback per model.

Finally, to determine the uncertainty associated with the regression coefficients in both observations and models, we calculate 5-95% confidence intervals using a two-tailed t-test that takes into account the variability in the data and provides confidence intervals that encompass both positive and negative deviations from the estimated regression coefficients.

We conduct a comparison between model results and observed internal variability feedbacks using TOA fluxes from the CERES instruments, Energy Balanced and Filled (EBAF) dataset updated to Ed4.1 (Loeb et al., 2018), and gridded temperature anomalies from HadCRUT version 5 (Morice et al., 2021). The comparison is performed during the overlapping period of the CERES-historical simulation (2001-2014). The objective is twofold: to identify models whose internal variability feedback differs from observations, providing valuable insights into their representativeness of forced climate feedbacks, and to investigate the lack of a robust relationships between net internal variability and net forced climate feedback despite the presence of strong relationships for the longwave and shortwave components, as reported in previous research (Uribe et al., 2022).

Furthermore, we systematically extend the historical period to assess the persistence of challenges in establishing a relationship between internal variability and forced net climate feedbacks in models over longer historical periods. Having identified a time period where the relationship emerges in models, we investigate the minimum conditions required to use observations and the model-based relationship to constrain the forced net climate feedback using ERA5 reanalysis data (Hersbach et al., 2020). We then use statistical time series modeling to align ERA5 reanalysis TOA fluxes with the combined datasets of CERES and the Earth Radiation Budget Experiment (ERBE) satellite records (Allan et al., 2014), in order to match the observational time length requirement. This methodology contributes to the derivation of an estimated emergent constraint on forced climate net feedback.

## 3 Results

### 3.1 Internal Variability and Forced Climate Feedbacks Relationship During the CERES Period

In order to be able to use internal variability feedbacks to constrain forced climate feedbacks there must exist a statistical relationship between these quantities. Indeed, during the overlapping years of the CERES satellite observations and historical CMIP6 simulations, there are high correlations between simulated internal variability and forced climate feedbacks (Figures 1a and 1b). Importantly, our results demonstrate a stronger correlation between internal variability and forced climate feedbacks compared to the findings reported by Uribe et al. (2022). We attribute this improvement in correlation strength to the combination of two factors: the inclusion of more models, allowing for a broader range of model representations and variations in internal variability to be captured, and the utilization of additional ensemble members, which enhances the robustness and representativeness of our internal variability feedbacks.

To assess the statistical significance of observed correlations between internal variability and forced climate feedbacks over this relatively short period, we conducted a Monte-Carlo permutation test. The null hypothesis assumed that no real relationship exists between internal variability and forced climate feedbacks, meaning any observed correlation would be due to random chance. To test this, we randomly permuted the feedback datasets, breaking the correspondence between models for internal variability and forced climate feedbacks (e.g., by pairing the internal variability feedback value from one model with the forced climate feedback value from another). We then recalculated the correlation coefficient using this permuted data. This process was repeated $10^5$ times, creating a null distribution of correlation coefficients that represents the range of correlation values we would expect if no actual relationship exists. Finally, we compared the observed correlation to this null distribution to estimate how often a correlation as large or larger than the observed one would occur by chance, providing a p-value as a measure of statistical significance. The results reveal a likelihood of less than 5% that the longwave and shortwave correlation coefficients would occur by chance alone (0.0% for both longwave and shortwave in CMIP, and 0.09% for longwave, 0.0% for shortwave in AMIP) and indicate a significant relationship between the strength of longwave and shortwave forced climate feedbacks and their corresponding internal variability feedback. By considering these relationships, observations have the potential to constrain and limit the uncertainties associated with forced climate feedbacks. The comparison of simulated and observed internal variability feedbacks reveals that models exhibiting moderate to strong negative longwave internal variability feedbacks, along with models featuring both weak negative and weak positive shortwave internal variability feedbacks, show more consistency with observed data (Figures 5a, 5b, 5d and 5e). Hence, even with the 14 year subset of the CERES dataset that overlaps with the climate model simulations we can constrain the feedback components.

In contrast, when examining the net feedback, we observe a breakdown in the statistical relationship (Figures 1c) as indicated by the relatively smaller correlation coefficient between internal variability ($\lambda_{it}$) and forced climate ($\lambda_{ab}$) net feedbacks ($r(\lambda_{it}, \lambda_{ab})$) and relatively larger probability of occurring by chance alone (0.73% for CMIP and 0.28% for AMIP, respectively). To explore the underlying causes for this relatively weaker relationship, we examine the computation of the correlation coefficient, which is defined as the covariance between internal variability and forced climate net feedbacks, normalized by

the product of their standard deviations $\left(r(\lambda_{it}, \lambda_{ab}) = \frac{Cov(\lambda_{it}, \lambda_{ab})}{\sigma(\lambda_{it})\sigma(\lambda_{ab})}\right)$. Utilizing the bilinearity of covariance and considering that net feedback is the aggregate of longwave $(\lambda_{lw})$ and shortwave $(\lambda_{sw})$ components, the correlation coefficient can be decomposed into its constituent parts as follows:

$$r(\lambda_{it}, \lambda_{ab}) = \frac{Cov(\lambda_{it\_lw}, \lambda_{ab\_lw})}{\sigma(\lambda_{it})\sigma(\lambda_{ab})} + \frac{Cov(\lambda_{it\_lw}, \lambda_{ab\_sw})}{\sigma(\lambda_{it})\sigma(\lambda_{ab})} + \frac{Cov(\lambda_{it\_sw}, \lambda_{ab\_sw})}{\sigma(\lambda_{it})\sigma(\lambda_{ab})} + \frac{Cov(\lambda_{it\_sw}, \lambda_{ab\_lw})}{\sigma(\lambda_{it})\sigma(\lambda_{ab})}. \quad (2)$$

This decomposition allows for a detailed examination of the contributions of longwave and shortwave feedback components to the overall correlation coefficient. Considering the observed correlations of individual longwave and shortwave feedbacks (Figures 1a and 1b), we expect positive contributions from the first and third terms in the right side of Eq. (2). Furthermore, alongside these positive correlations, there is a tendency for longwave and shortwave internal variability feedbacks to counteract each other ($r(\lambda_{it\_lw}, \lambda_{it\_sw}) = -0.66$ and $r(\lambda_{it\_lw}, \lambda_{it\_sw}) = -0.62$ for CMIP and AMIP, respectively). This suggests potential inverse relationships between internal variability feedbacks and its corresponding counterpart forced climate feedback, consequently leading to anticipated negative contributions from the second and last term in Eq. (2). In the CMIP dataset, the terms have values of 0.43, -0.83, 1.57 and -0.72, while in the AMIP dataset, they are 0.33, -0.63, 1.46 and -0.63, respectively. These values show that the relatively weaker covariance between longwave internal variability and longwave forced climate feedbacks contributes to the overall weak relationship observed in the net feedbacks from both CMIP and AMIP datasets.

## 3.2 Emergence of the Relationship Between Net Internal Variability and Net Forced Climate Feedbacks

Whereas the relationships between longwave and shortwave internal variability and forced climate feedbacks remain robust during the CERES period, a comparable strength of the relationship is notably absent for the net feedback component. To determine the potential existence of such a relationship between internal variability and forced climate net feedbacks in CMIP6 models, we extend our analysis to longer time periods, surpassing the length of the CERES observational record.

To this end, we calculate the correlation coefficient components between internal variability and forced climate net feedbacks (Eq. (2)) across various time-window sizes in CMIP6 simulations (Figure 2a). Additionally, we determine the corresponding correlation coefficient and p-value through a hypothesis test, where the null hypothesis posits independence and no correlation between internal variability and forced climate net feedbacks. (Figure 2b). The calculation starts by using a 14-year historical simulation time window (2001-2014), and then extending the window by one year at a time until reaching the initial year of 1850 (1850-2014). Here, we use just historical coupled simulations since the time length of historical atmosphere-only simulations is shorter (1979-2014). Additionally, note that the historical simulations span 165 years; however, to eliminate the influence of volcanic eruptions (Krakatoa in 1883-1884, Agung in 1963-1964, El Chichon in 1982-1983, and Pinatubo in 1991-1992) that perturbed the TOA fluxes in ways unrelated to internal variability, we excluded the corresponding years from the time series. As a result, the length of the record was reduced to 157 years.

The results reveal three key points. First, the significance of the relationship for the net feedback depends on the chosen time span for estimating internal variability net feedback (Figure 2b). Secondly, this dependency eventually weakens, and notably

robust and significant relationships between internal variability and forced climate net feedbacks emerge. Thirdly, while the weaker covariance between longwave internal variability and longwave forced climate feedbacks primarily contributes to the weak correlation during the CERES period, the covariance between shortwave internal variability and shortwave forced climate feedbacks is responsible for bolstering the relationship between net feedbacks when extending the data record (Figure 2a). Accounting for the years corresponding that are excluded from the analysis due to volcanic eruptions, this period represents an effective record length of 51 years. Consequently, if observational records were long enough, they could aid in constraining the forced climate net feedback. However, due to the limited availability of observational data (Figure 2), obtaining a purely observational estimate of net internal variability feedback to constrain net forced climate feedbacks is not feasible at this point in time.

Combining the Earth Radiation Budget Experiment (ERBE) satellite record, which started in 1985, with the existing CERES data (Allan et al., 2014) extends the available observational period to 37 years. Thus, about 14 years of additional continuous observational data, undisturbed by volcanoes, would need to be collected to reach a total of 51 years. To assess the likelihood of a significant relationship between internal variability and forced climate net feedbacks in CMIP6 coupled models over a 51-year window other than 1958-2014, we calculate the frequency distribution of p-values between them for all 51-year non-volcano consecutive periods between 1850 and 2014 (Figure 2c). The analysis reveals a high frequency of p-values below one percent, indicating a strong likelihood that with an additional 14 years of satellite data from the end of 2023, the use of purely observed internal variability net feedback as a constraint on net forced climate feedback would be feasible and reliable.

So far we have demonstrated the emergence of relationships between forced climate and internal variability net feedbacks. The most recent potential relationship, between 1958 and 2014, unfortunately has missing periods of observational data. While future efforts may yield enough observations to cover the same time span as the emergent relationship, historical simulations end in 2014, and if they are not updated alongside new data collection, this poses the question of whether an emergent relationship from one period can be applied to observations from another period, as it has been suggested in other studies (e.g. Dessler and Forster (2018)). Alternatively, there is the consideration of using the emergent relationship from 1958 to 2014 with the available 14 years of CERES observations; however, a key question remains: are 14 years of internal variability feedback observations significantly representative of a 51-year span? We address these questions in the following section.

### 3.3 The Minimal Requirements for A Reliable Emergent Constraint Estimation

We now examine the question concerning the applicability of emergent relationships between forced climate and internal variability net feedbacks to different periods. Our analysis addresses the practicality of applying the emergent relationship between 1958 and 2014 to periods without overlapping observations and examines the comparability between 14-year internal variability feedback observations and those spanning 51 years. Through these investigations, we aim to uncover the essential requirements necessary to establish a statistically robust emergent constraint on forced net climate feedback.

To assess the transferability of the emergent relationship from the 1958-2014 period to observation periods of the equivalent duration but from different years, we employ a proof by contrapositive approach. We hypothesize that observations from one

51-year period can be used with the model relationship between $\lambda_{it}$ from a different 51-year period and $\lambda_{ab}$ to derive the emergent constraint. If this hypothesis is valid, model relationships derived from different 51-year periods for $\lambda_{it}$ should be statistically similar. Therefore, we compute the slopes and intercepts of all potential 51-year model relationships from 1850 to 2014 and compare them with the confidence intervals of their means. The analysis results (Figures 3a and 3b) indicate significant discrepancies in both the slope and intercept of the 51-year emergent relationships when compared to their mean

confidence intervals. However, there is some degree of similarity, with slopes and intercepts from adjacent time periods sometimes showing close values, yet this trend is not consistent across the entire analyzed period. Thus, the acceptable degree of mismatch between time periods needed to calculate internal variability from observations and models for deriving emergent constraints on forced climate feedbacks depends on the specific periods being compared. These discrepancies underscore the risk of using emergent relationships and observations from 51-year periods that originate from different years. The emergent

relationship in one 51-year period may not align significantly with that of another period, potentially misleading the interpretation of forced climate net feedbacks and undermining the reliability of emergent constraints.

    In addressing the second question, regarding the use of observed internal variability feedback from the 14-year period (2001-2014) with the emergent relationship from the period 1958-2014 to constrain forced climate net feedback, the hypothesis is that, if valid, there should be no significant differences between any 14-year internal variability net feedback within the 1958-

2014 period and the corresponding net feedback of this period. Using ERA5 reanalysis data, all conceivable 14-year internal variability net feedbacks were calculated and compared with the 5-95% confidence interval of the internal variability net feedback of these 51 years (Figure 3c). The results indicate significant differences, negating the hypothesis. Consequently, using the 14-year CERES observations (2001-2014) in conjunction with the emergent relationship (1958-2014) to constrain forced climate feedback could yield erroneous results, as a given 14-year period may have observed internal variability that is

significantly different from another.

    Whereas the above analysis highlights that a 14-year period will not yield a statistically significant internal variability net feedback estimation, we next ask how long an observation period is required to yield a significant estimate. This minimal period length (n years) should show significant agreement between all conceivable internal variability net feedback within the 51-year timeframe and that estimated from the entire period. To investigate the existence of such a period, we calculated

all conceivable internal variability net feedbacks over different n-year periods within the 1958-2014 time frame. We then determined the percentage of internal variability net feedback that deviated from the confidence intervals of the total period internal variability net feedback for each n (Figure 3d). The results indicate that for a 40-year period, or longer, the internal variability net feedback estimate is not significantly different from that of the 1958-2014 period. This suggests that after merging the CERES (2001-2014) and ERBE (1985-2000) datasets, only 12 non-volcanic years (1971-1984) would be required

to meet the minimum time requirement. It is worth noting that the time span from 1971 to 2014 is 44 years, but by excluding volcanic eruption years (El Chichon and Pinatubo), the effective time span is reduced to 40 years. In the following section,

we leverage ERA5 reanalysis and statistical time series modeling to satisfy the above conditions and construct an emergent constraint on forced net climate feedback.

## 3.4 Extending the Observational Record with Reanalysis Data

As we have shown in the previous section, about 40 years of observations within the period where the relationship in models emerges is the minimum requirement to establish an emergent constraint on the forced net climate feedback. The combined CERES-ERBE dataset (1985-2014) contains 28 non-volcanic years, leaving 12 non-volcanic years missing to meet the required time frame. In this section, we turn to ERA5 reanalysis data as a proxy for satellite observations to fill the observational gap and extend the accessible time span (1971-2014) for estimating internal variability net feedbacks. Before proceeding, however, a detailed examination of the ERA5 reanalysis is essential, since in a reanalysis the radiative fluxes are not directly constrained by observations, but are derived from calculations based on the atmospheric state. While the distribution of temperature and water vapor is relatively well-constrained in a reanalysis, the representation of clouds is less constrained, resulting in radiative fluxes that may be less consistent with observations than other variables.

To assess the extent to which ERA5 captures the patterns and variations present in the observational data, we conduct a comparison of the global weighted mean TOA fluxes from ERA5 with those obtained from the combined CERES-ERBE dataset over the overlapping period of 1985-2019 (Figure 4). The high correlation coefficients obtained from the comparison indicate the reanalysis effectively represents the variations affecting the TOA fluxes (Figure 4a). However, a closer examination reveals that ERA5 has a distinct error pattern (Figure 4a). Broadly speaking, this pattern indicates a systematic underestimation of longwave fluxes and an overestimation of shortwave fluxes. Interestingly, these individual errors appear to cancel each other out, leading to a net flux that is in relatively good agreement with the observed data.

Furthermore, a closer look at these errors reveals several predictable time series characteristics, such as temporal autocorrelation, seasonality, and trends (not shown). Notably, these features are more pronounced in the shortwave and net TOA fluxes, while they are less apparent in the longwave. Such characteristics highlight a potential of utilizing time-series statistical modeling techniques for predicting errors in the reanalysis that then in turn can be used to fill periods lacking observations. Guided by the recognized characteristics, we chose a Seasonal Autoregressive Integrated Moving Average (SARIMA) model (Box et al., 2008).

To build the model, we divided the time series into a training set (January 1987 to December 2019), allowing the model to learn patterns and relationships within the data (Figure 4a). We then evaluate the model's performance using a test over an independent time period not seen during training (January 1985 to December 1986). This test ensures that the model can accurately predict TOA fluxes for unseen periods. Note that, given the ability of the SARIMA model to forecast, a reversal in the time series is essential to address the need for completing errors in the past. The test shows excellent performance in predicting shortwave and net TOA fluxes, while its accuracy is slightly lower for longwave (Figure 4b). This discrepancy may arise from the less pronounced error patterns in longwave. Nevertheless, a majority of the test observations fall within the confidence intervals of the model results (the model's 5-95 percent confidence intervals encompass roughly 75 percent of

the longwave, 100 percent of the shortwave, and 87.5 percent of the net test datasets), providing confidence in its predictive capabilities. Using the validated SARIMA model, we forecast errors from 1971 to 1984, enabling the adjustment of ERA5 TOA fluxes (Figure 4c). This ensures a seamless 40-year period representative of the required 51 years and allows us to establish a constrained emergent relationship in models spanning the period 1958-2014.

Now equipped with an emergent relationship in models (1958-2014) and a 40-year adjusted observation set (1971-2014) from ERA5-CERES-ERBE, we can establish an emergent constraint on forced climate feedbacks (Figures 1d, 1e and 1f). To that end we utilize a Monte Carlo simulation approach. We start by generating a predictor variable dataset sampled from a truncated normal distribution, based on the 40-year adjusted ERA5-CERES-ERBE observations and their associated confidence intervals. For each sampled predictor value, we compute the corresponding confidence interval for the predicted forced climate

feedbacks using a linear model ($\lambda_{ab} = m\lambda_{it} + b$) that captures the emergent relationship between forced climate and internal variability feedbacks. While OLS regression can derive this linear model, it may underestimate the slope due to regression dilution caused by uncertainties in the predictors. To account for these uncertainties in the predictor, we also employ Orthogonal Distance Regression (ODR). The comparison of the results from both regression methods (see Table 2) shows minimal impact of regression dilution on the estimated slope when using OLS, leading us to choose OLS for its simplicity. Subsequently,

we sample from a truncated normal distribution based on the derived predicted confidence intervals to address prediction uncertainties, resulting in a new dataset of predicted forced climate feedback values. The emergent constraint on forced climate feedbacks is then characterized by the probability density function of this new dataset (insets in Figures 1d, 1e, and 1f).

    Having explained the procedure for the emergent constraint, we now focus on the discernible results it provides. A comparison of the probability distribution medians shows that, for longwave, the emergent constraint (-1.74 Wm$^{-2}$ K$^{-1}$) closely

matches the model median (-1.79 Wm$^{-2}$ K$^{-1}$) (Figure 1d). For shortwave, however, the emergent constraint suggests a less positive forced climate feedback (0.05 Wm$^{-2}$ K$^{-1}$) compared to the model median (0.59 Wm$^{-2}$ K$^{-1}$), indicating a reduced shortwave TOA response to global warming (Figure 1e). As a result, the emergent constraint indicates a more negative forced net climate feedback (-1.52 Wm$^{-2}$ K$^{-1}$) compared to the model median (-1.12 Wm$^{-2}$ K$^{-1}$), reflecting the diminished positive shortwave feedback (Figure 1e). Consequently, based on a radiative forcing for a doubling of carbon dioxide of 3.93 Wm$^{-2}$

($\pm$ 0.47 Wm$^{-2}$ with 5-95% confidence intervals) (Forster et al., 2021), our emergent constraint on forced net climate feedback suggests an ECS with a median of 2.59 K, and 5–95% confidence intervals spanning from 1.95 K to 3.12 K, which is somewhat lower than the model median of 3.09 K. As an alternative to using the IPCC AR6 radiative forcing estimate, the radiative forcing can be retrieved from the model set by taking the y-intercept of the regression between TOA fluxes anomalies and surface temperature anomalies, then dividing by 2.1 (Gregory et al., 2004; Meinshausen et al., 2020). Here, the anoma-

lies are computed as the difference between the abrupt4XCO2 and piControl 150 year experiments. This approach yields an ECS with a median value of 2.31 K and 5–95% confidence intervals ranging from 1.74 K to 2.75 K. These ECS estimates, although consistent with previous research, lie towards the lower end of the reported ranges (Dessler and Forster, 2018; Ceppi and Nowack, 2021; Sherwood et al., 2020; Forster et al., 2021). Lastly, the emergent constraint outlined above, along with a comparison between simulated and adjusted ERA5-CERES-ERBE observational internal variability feedback, indicates that

models exhibiting moderate negative longwave, weakly positive shortwave and strongly negative net feedbacks are in better agreement with observations (Figures 5g, 5h and 5i).

However, three critical factors must be considered. First, the method used here assumes that the relationship between internal variability and forced climate feedbacks identified in models also applies to the real world. Yet, those models often fail to accurately replicate observed sea surface temperature trend patterns (e.g., Armour et al., 2024; Wills et al., 2022; Seager et al.,
2022), which can influence forced climate feedbacks. This mismatch may result in deviations between model-based and real-world relationships between internal variability and forced climate feedbacks. Second, uncertainties in the model emergent relationship within the adjusted ERA5-CERES-ERBE period, particularly because most models simulate internal variability net feedback values significantly outside the observed range (Figure 1f) reduces confidence in the emergent constraint. Finally, the emergent constraint is derived from a 40-year adjusted ERA5-CERES-ERBE period, which implies reduced confidence
compared to a purely observational 51-year dataset. Given these limitations, it is important to interpret the presented emergent constraint not as strong evidence, but rather as a prediction of the potential insights that could be gained with little more than a decade of additional observations.

## 4 Conclusions

A study of the relationships between internal variability and forced climate feedbacks in models of the sixth generation of
340 the Coupled Model Intercomparison Project is presented. Consistent with previous research (Uribe et al., 2022) we find evidence indicating that the strength of longwave and shortwave forced climate feedback is related to their internal variability feedback during the period of overlap between CERES data and historical simulations. Moreover, our results indicate that the inclusion of additional ensemble members in the estimation of internal variability feedbacks improves their robustness and representativeness, thereby strengthening the relationship between internal variability and forced climate feedbacks.
Nevertheless, when the longwave and shortwave feedback components are combined to estimate the net feedback, the relationship breaks down. We showed that the relatively weaker relationship between longwave internal variability and longwave forced climate feedbacks played a significant role in the reduced relationship between internal variability and forced climate net feedbacks during the CERES period.

To determine the relationship between internal variability and forced net climate feedback in CMIP6 coupled simulations,
we extend the analysis to encompass time periods beyond the constraints of CERES observations. The results indicate robust and statistically significant relationships within the model set, with increased statistical stability for a period of at least half a century, excluding years with volcanic eruptions (1958-2014). The improvement observed in the relationship stems from a strengthening of the correlation between shortwave internal variability and shortwave forced climate feedbacks when extending the period. This increase compensates for the relatively weaker correlation between longwave internal variability and longwave
forced climate feedback. Given that the CERES record begins in 2000, our analysis suggests that it would be necessary to wait until approximately 2051 to accumulate the necessary satellite observations for the use of purely observational data in

constraining forced climate feedbacks. However, a possible solution to reduce this waiting time is to combine the CERES record with the ERBE satellite record, which could potentially reduce the time horizon to the mid-2030s.

We then find that a timeframe of about 40 years, within the period wherein the model internal variability net feedback demonstrates a relatively robust relationship with the model forced net climate feedback, represents the essential criteria for employing observations to establish an emergent constraint on forced net climate feedback. Leveraging the 1985-2014 CERES-ERBE dataset, which includes 28 non-volcanic years, we employ time series modeling to adjust ERA5 reanalysis data. This adjusted reanalysis data record serves as a surrogate for satellite observations, with the goal of filling the observational gap and extending the available time frame for estimating observed internal variability net feedback (1971-2014).

Using this extended observation set (1971-2014) in conjunction with the emergent statistical relationship in models (1958-2014), we derive an estimated emergent constraint on forced climate feedbacks. This constraint manifests as a reduction in the uncertainty associated with forced climate feedbacks, revealing a reduced forced climate shortwave feedback, a more pronounced negative forced climate net feedback, and consequently, a lower Equilibrium Climate Sensitivity relative to the CMIP6 model distribution. In particular, it is highlighted that models with moderate negative longwave, weakly positive shortwave, and strongly negative net feedbacks are more consistent with observations.

However, a few final notes of caution are warranted. First, at the time scales we are examining, for example some bio-geophysical and biogeochemical feedbacks may not be active (e.g., changes in methane, aerosols, ozone, or vegetation), and several of the models used also do not incorporate them. Consequently, this absence may lead to an underestimation of ECS to some extent. Second, it remains uncertain whether the identified model relationship between internal variability and forced climate feedbacks reflects real-world conditions. Third, uncertainties in the model emergent relationship within the adjusted ERA5-CERES-ERBE data limit confidence in the emergent constraint results. Furthermore, it is important to recognize that these emergent constraints are derived from a 40-year adjusted ERA5-CERES-ERBE period, indicating a reduced level of confidence compared to a purely observational 51-year dataset, highlighting the need for long term continuous monitoring of Earth's radiation budget. Given these limitations, it is advisable to interpret the emergent constraint results carefully, understanding that they serve as indicative illustrations rather than strictly observational evidence.

*Data availability.* The CERES EBAF-TOA Ed4.1, and combined CERES-ERBE data sets used for estimating observed TOA fluxes are available at the NASA Langley Research Center via https://ceres.larc.nasa.gov/data/ and University of Reading via https://www.met.reading.ac.uk/%7Esgs02rpa/research/DEEP-C/GRL/. The Gridded temperature anomalies HadCRUT(5) data set used to estimate observed internal variability feedbacks can be obtained from the Met Office Hadley Centre observations datasets at https://www.metoffice.gov.uk/hadobs/hadcrut5/. Used CMIP6 models can be found in Table S1 and are available from ESGF at https://esgf-node.llnl.gov. ERA5 reanalysis data can be downloaded from the Copernicus Climate Change Service in https://doi.org/10.24381/cds.f17050d7.

*Author contributions.* The project idea originated from collaborative discussions among all authors. AU conducted the data analysis and drafted the manuscript. All authors contributed to discussions, provided edits, and reviewed the manuscript for accuracy and coherence.

*Competing interests.* The authors declare that they have no conflict of interest.

*Acknowledgements.* The study was supported by the European Research Council project highECS (grant no. 770765), the Swedish Research Council (grant no. 2018-04274), the European Union's Horizon 2020 program projects CONSTRAIN (grant no. 820829) and NextGEMS (grant no. 101003470).

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

Table 1. CMIP6 models used to calculate forced climate feedbacks and internal variability feedbacks. The availability of model realizations for estimating internal variability feedbacks is indicated by "X".

| | r1i1p1f1 | | r2i1p1f1 | | r3i1p1f1 | | r4i1p1f1 | | r5i1p1f1 | |
|---|---|---|---|---|---|---|---|---|---|---|
| | CMIP | AMIP | CMIP | AMIP | CMIP | AMIP | CMIP | AMIP | CMIP | AMIP |
| ACCESS-CM2 | X | X | X | X | X | X | X | X | X | |
| ACCESS-ESM1-5 | X | X | X | X | X | X | X | X | X | X |
| AWI-CM-1-1-MR | X | | X | | X | | X | | X | |
| BCC-CSM2-MR | X | X | | X | | X | | | | |
| BCC-ESM1 | X | X | X | X | X | X | | | | |
| CAMS-CSM1-0 | X | X | X | X | | X | | | | |
| CanESM5 | X | X | X | X | X | | X | | X | |
| CESM2 | X | X | X | X | X | X | X | X | X | X |
| CESM2-FV2 | X | X | X | X | X | X | | | | |
| CESM2-WACCM | X | X | X | X | X | X | | | | |
| CESM2-WACCM-FV2 | X | X | X | X | X | X | | | | |
| CMCC-CM2-SR5 | X | X | | | | | | | | |
| FGOALS-g3 | X | X | X | X | X | X | X | X | | X |
| GISS-E2-1-G | X | | X | X | X | X | X | X | X | X |
| GISS-E2-1-H | X | | X | | X | | X | | X | |
| ICON-ESM-LR | X | X | X | | X | | X | | X | |
| IITM-ESM | X | X | | | | | | | | |
| MIROC6 | X | X | X | X | X | X | X | X | X | X |
| MPI-ESM-1-2-HAM | X | X | X | X | X | X | | | | |
| MPI-ESM1-2-HR | X | X | X | X | X | X | X | | X | |
| MPI-ESM1-2-LR | X | X | X | X | X | X | X | | X | |
| MRI-ESM2-0 | X | X | X | X | X | X | X | | X | |
| NESM3 | X | X | X | X | X | X | X | X | X | X |
| NorESM2-MM | X | | X | | X | | | | | |
| SAM0-UNICON | X | X | | | | | | | | |

**Table 2.** Forced climate and internal variability feedbacks regression coefficients depending on the choice of regression method.

| | Longwave | | Shortwave | | Net | |
|---|---|---|---|---|---|---|
| | OLS | ODR | OLS | ODR | OLS | ODR |
| Slope | 0.57 | 0.57 | 0.74 | 0.74 | 0.57 | 0.62 |
| Intercept [$Wm^{-2}K^{-1}$] | -0.77 | -0.76 | -0.26 | -0.26 | -0.82 | -0.80 |

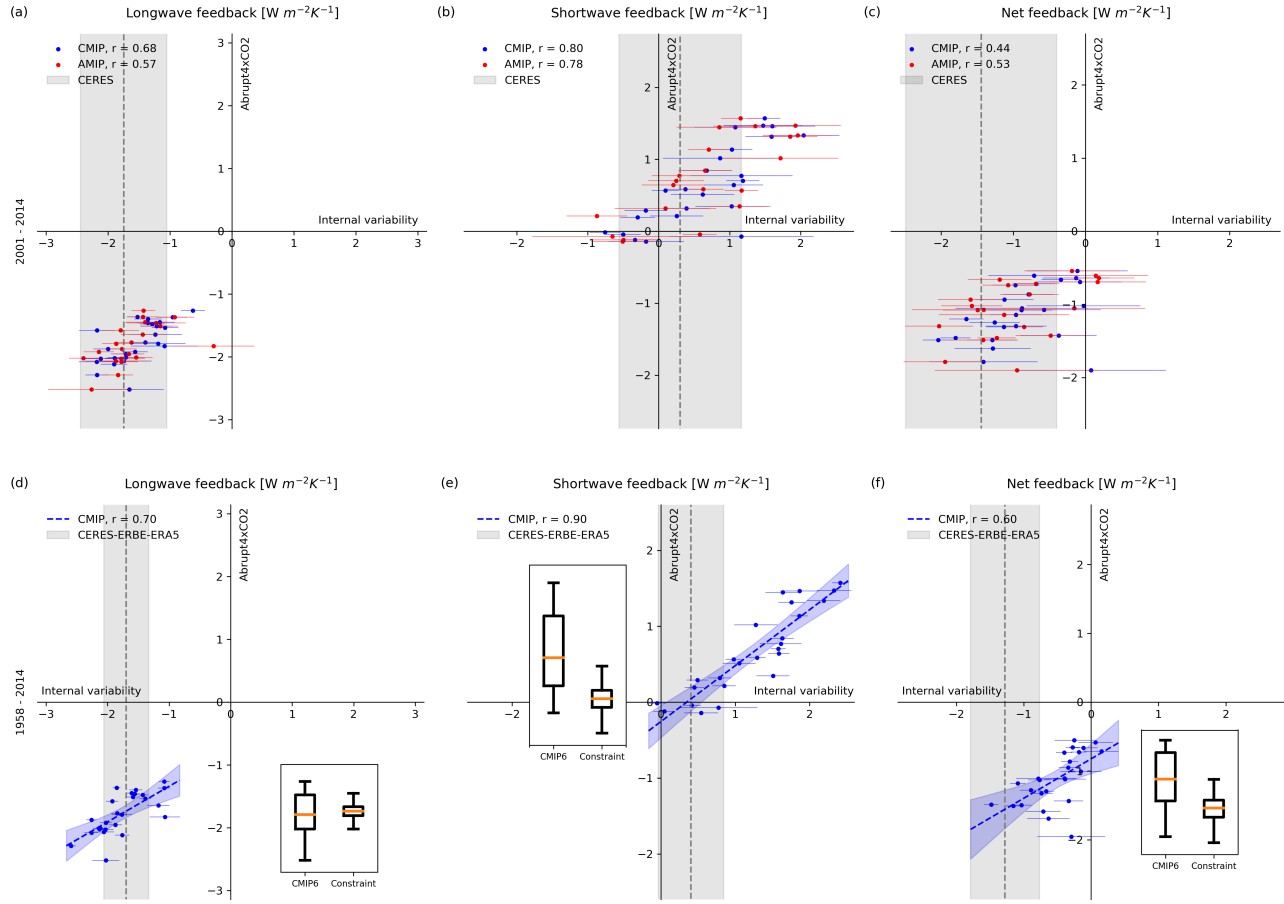

**Figure 1.** Abruptly quadrupled $CO_2$ increases versus internal variability feedbacks for longwave (left), shortwave (center), and net (right) over CERES 2001-2014 ((a), (b), (c)) and 1958-2014 ((d), (e), (f)) periods. Legends include the correlation coefficient, $r$, between abrupt and internal variability feedbacks for coupled (blue) and atmosphere-only (red) simulations. Grey shading and horizontal lines extend from 5 to 95 percent confidence intervals of the observed/ERA5-CERES-ERBE and individual model internal variability feedback, respectively. The shaded blue region in figures (d), (e), and (f) represents the 5 to 95% confidence intervals on the regression line, which is displayed as a dashed line. Additionally, the insets display boxplots for CMIP6 and constrained forced climate feedbacks.

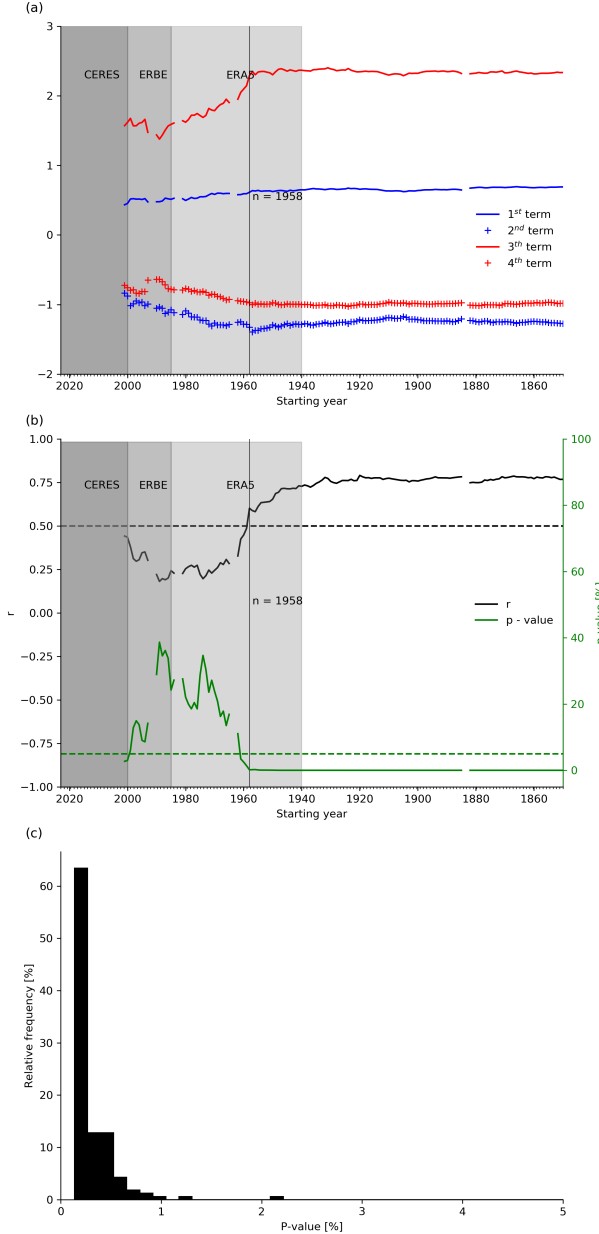

**Figure 2.** (a) Shows the first (blue line), second (blue crosses), third (red line), and fourth (red crosses) contributors to the correlation coefficient between internal variability and forced climate net feedbacks (Eq (2)). (b) Ilustrates the correlation coefficient (black line) and p-value (green line) between internal variability and forced climate net feedbacks, with horizontal dashed lines indicating a correlation of 0.5 and a p-value of 5%. The values are computed between the net internal variability feedback estimated for the Starting year to 2014 time window and net forced climate feedback. The vertical line in figures (a) and (b) marks the initial year when the relationship approximately stabilizes, while the shaded grey regions represent the CERES (2000-present), ERBE (1985-2000), and ERA5 (1940-present) periods. (c) Depicts the frequency distribution of p-values from de-correlation tests between internal variability and forced climate net feedbacks in CMIP6 coupled models over all 51-year windows between 1850 and 2014.

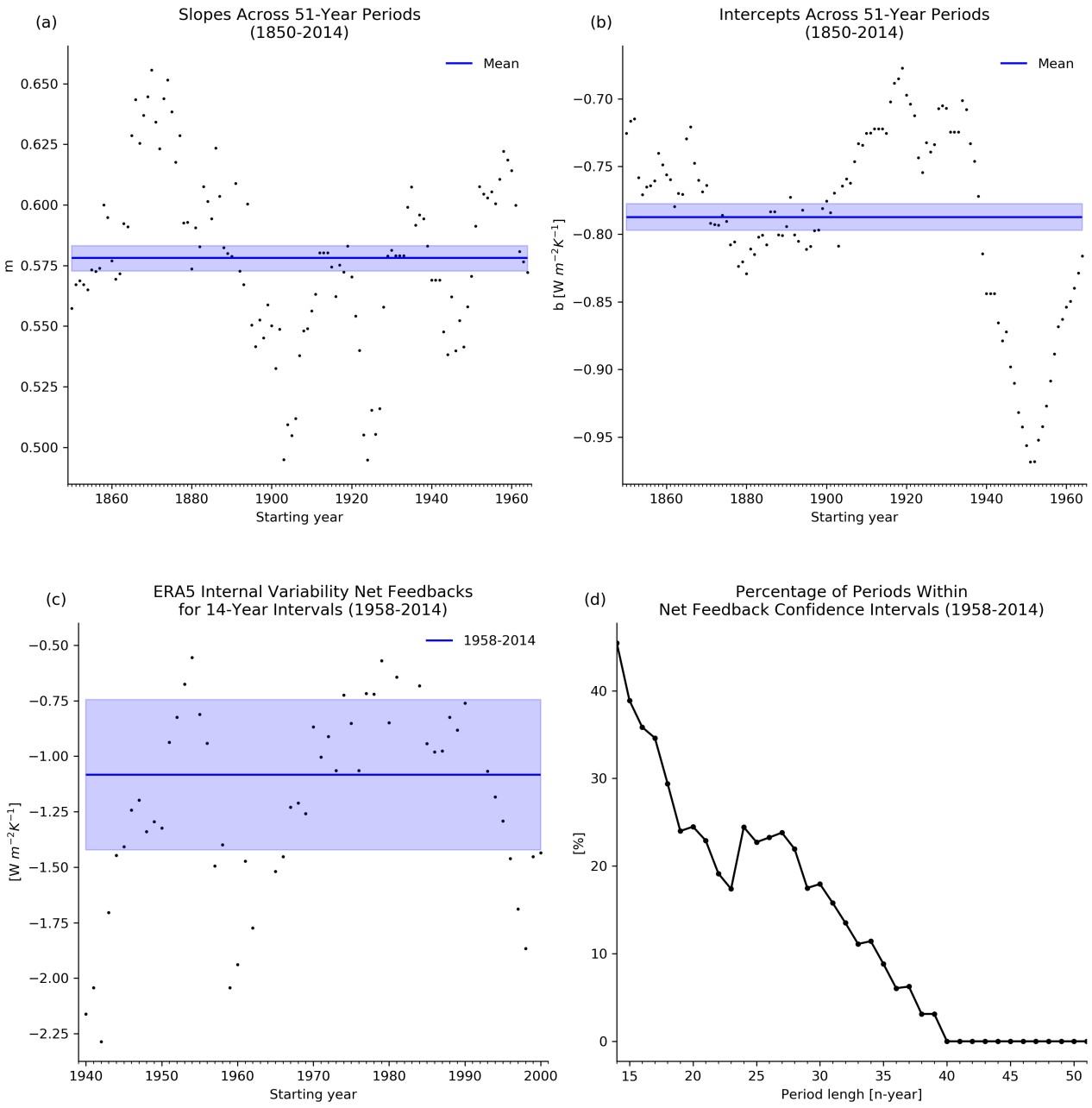

**Figure 3.** Regression slopes (a) and intercepts (b) of the linear relationship between internal variability and forced climate net feedbacks for all potential 51-year continuous periods spanning from 1850 to 2014 in CMIP6 coupled simulations and the 5-95% confidence intervals on their mean (blue). (c) ERA5 internal variability net feedbacks for every possible 14-year interval spanning from 1958 to 2014. The shaded blue intervals represent the 5-95% confidence intervals across the entire time span. (d) Percentage of periods within the confidence intervals of internal variability net feedbacks for the 1958-2014 period relative to the period length.

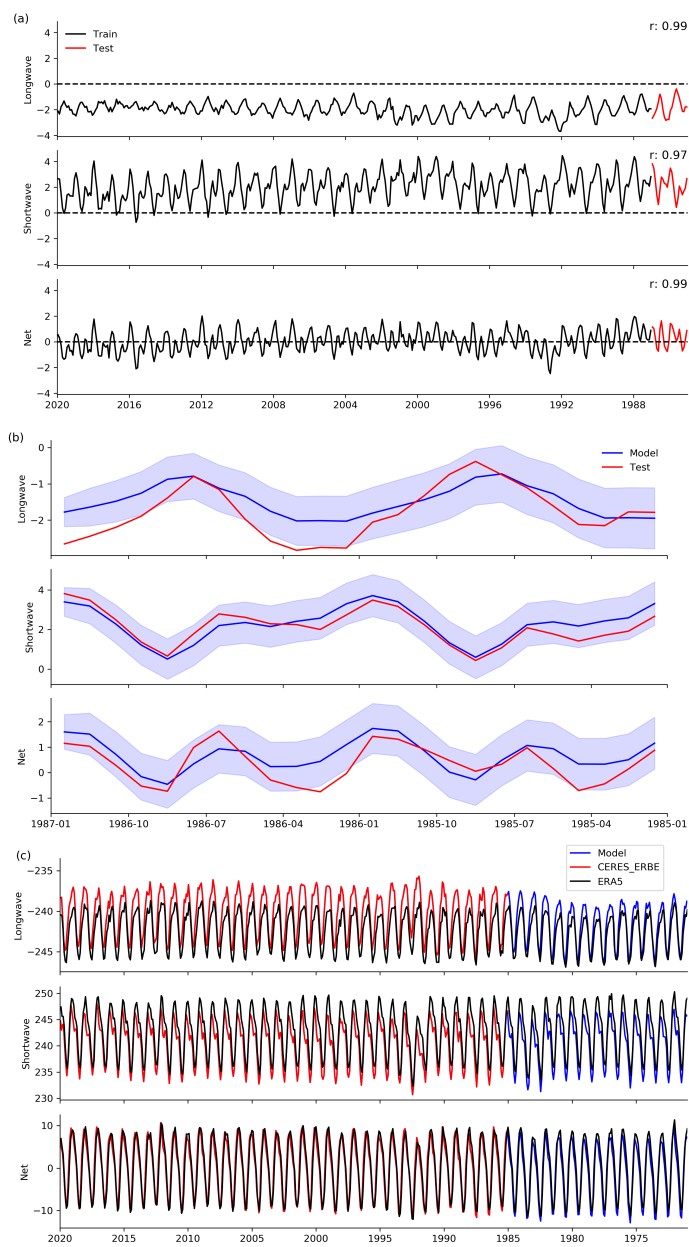

**Figure 4.** (a) Shows the bias in global mean ERA5 TOA fluxes (ERA5 - CERES-ERBE) from 1985 to 2019, with the black and red lines indicating the training and testing periods for the SARIMA model, respectively. Legends provide the correlation coefficient between ERA5 and CERES-ERBE. (b) Illustrates the bias in global mean ERA5 TOA fluxes during the testing period (red) and the SARIMA model results with its 5-95% confidence intervals (blue). (c) Depicts TOA fluxes from CERES-ERBE spanning 1985 to 2019 (red), ERA5 spanning 1971 to 2019 (black), and SARIMA adjusted ERA5 spanning 1985 to 2019 (blue).

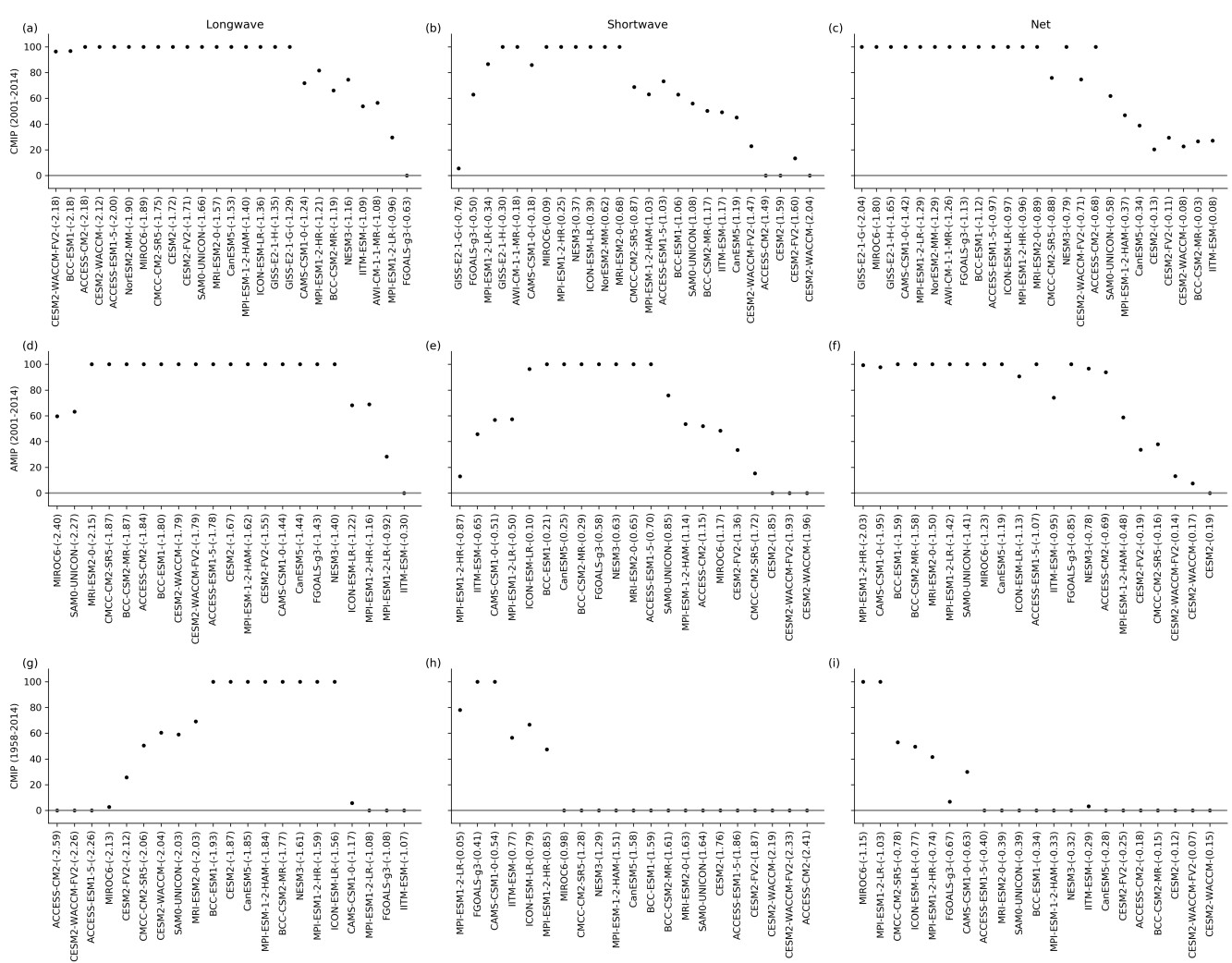

**Figure 5.** Probability of longwave (left) shortwave (center) and net (right) internal variability feedbacks from models falling within observed uncertainty ranges. The top and center panels show CMIP and AMIP simulations, respectively, alongside CERES observations for the period 2001-2014, while the bottom panel illustrates CMIP simulations and the combined CERES-ERBE-ERA5 covering the broader timeframe from 1958 to 2014. The x-axis organizes models in ascending order based on internal variability feedback values, with the corresponding values enclosed in parentheses.

