# Peer review of "Constraining net long term climate feedback from satellite observed internal variability possible by mid 2030s"

_EGUsphere, 2024_

## Author Comment (AC1)

**Response to Referee Comments on "Constraining net long term climate feedback from satellite observed internal variability possible by mid 2030s"**

We appreciate the referees' constructive feedbacks. Below, we provide our detailed responses and the modifications made in response to their comments and suggestions.

- **Referee 1.**

Using CMIP6 model simulations, the authors derive an emergent constraint that relates feedback from internal variability (IV) to forced feedback. They show that there are statistically significant relationships across models between components of IV feedback and forced feedback: a strong relationship for SW, a weaker relationship for LW, and an even weaker, but still significant and meaningful relationship for the net feedback. Using this relationship and combining it with observed internal variability, they show that more observations are needed in order to use this finding to actually constrain ECS. As an alternative to waiting for more satellite data, the authors extend the satellite record back in time by applying a correction model to reanalysis radiative fluxes, and thus constrain ECS to 2.5 K [90 % CI: 2.14 – 3.07 K], which is lower than current estimates from models, the IPCC, or Sherwood et al. 2020. Future satellite observations could be integrated into their method to update the estimates, and the authors quantify the quality of the constraint as a function of the number of observed years. The paper is well-written, well-presented, clearly states its goal and provides evidence to support the claims, guiding the reader through the argumentation. The statistical methods are sound and used in an appropriate way. While I do have a long list of comments and questions, I want to stress that I enjoyed reading the paper and consider it a beneficial addition to the research on feedbacks and climate sensitivity. I have one main point to raise in criticism of this paper, which I will present in the following.

My main comment can be summarized as "What about the pattern effect?"

From the methods section, I understand that the feedback parameters are calculated as differential feedback parameters (referring to Rugenstein and Armour 2021, `https://doi.org/10.1029/2021GL092983`, please confirm if this interpretation is correct). All feedback parameters are estimated as the slope of N(T). For $\lambda_{ab}$, which time period is used for the regression? The full 150 years? We know that $\lambda_{ab}$ changes considerably over time, both over the 150 years period (which is accounted for if the full 150 years are used for the regression), but also after this (e.g. Rugenstein et al. 2020, `https://doi.org/10.1029/2019GL083898`). According to that paper, ECS estimated from the 150-year span is an underestimate of the true ECS by 17 % in models. Would this affect the ECS estimate that the paper gives?

Further uncertainties may arise when leaving the model world. The historical simulations which are used to compute $\lambda_{it}$, are not capable of reproducing the observed SST patterns (e.g. Wills et al. 2022, `https://doi.org/10.1029/2022GL100011`). It is currently debated if the observed pattern of strong Western Pacific warming will continue or switch to stronger warming in the Eastern Pacific. This uncertainty implies enormous uncertainty for ECS (Alessi and Rugenstein 2023, `https://doi.org/10.1029/2023GL105795`). The point that I'm trying to make with these explanations is that it may very well be that the connection between $\lambda_{it}$ and $\lambda_{ab}$ is very different in the real world and models. While models produce El-Nino like patterns both in the present and future, the real world has warmed more La-Nina like until now, and we don't know how it will continue. Since these patterns are tightly linked to $\lambda_{ab}$, the model results may not be applicable to the real world. This would be a major problem for the emergent constraint that the paper develops, because an implicit assumption of the emergent constraint approach is that the statistical relationship that is found in the models is applicable to reality.

I would like to ask the authors to discuss this uncertainty. In particular, do you think it affects the ECS range that is determined? If yes, how? If no, why not? If the authors agree that this could add substantial uncertainty, I propose mentioning this also in the last part of the abstract, which currently suggests that all uncertainties (except for the biogeochemical feedback) are accounted for in the 5 – 95 % CI.

**Your interpretation regarding the feedback parameters calculated as differential feedback is accurate, and we derived them using 150 years of simulations.**

**While we recognize that ECS estimates derived from 150-year experiments may underestimate the true ECS due to time-dependent feedbacks, as shown by Rugenstein et al. (2020), it's equally important to consider that ECS estimates from 4XCO2 experiments frequently overestimate ECS compared to 2XCO2 experiments due to nonlogarithmic forcing, feedback CO2 dependence, and feedback temperature dependence, as demonstrated by Bloch-Johnson et al. 2021. Since the ongoing debate on the interaction of these two effects lies beyond the scope of our study, we chose the standard method as it represents the most straightforward approach and serves as a basis for comparison and analysis. This observation has been incorporated**

into the Materials and Methods section, line 85.

Indeed, an essential assumption in our study is that the relationship between $\lambda_{it}$ and $\lambda_{ab}$ observed in models also holds true in the real world. You rightly highlight that, so far, the observed warming pattern may differ from model projections, with the real world displaying more La Niña-like trends while models generally exhibit El Niño-like patterns. This discrepancy introduces significant uncertainty for our $\lambda_{ab}$ and ECS estimates. Given the importance of this caveat, we have include this potential discrepancy between models and observations in the abstract, discussion section and final notes of caution in our manuscript.

In addition, I have other comments:

- l. 12, 21, 335 – 337: What biogeochemical processes does this refer to? Can you specify? I wonder if they are relevant for ECS, as the carbon-cycle does not matter for this concept of fixed CO2 concentration, and vegetation changes are not included in the definition of ECS.

  While several definitions of ECS do not include biogeochemical processes, we align with the IPCC's definition as detailed in Forster et al. 2021:

  "Feedbacks in the Earth system are numerous, and it can be helpful to categorize them into three groups: (i) physical feedbacks; (ii) biogeophysical and biogeochemical feedbacks; and (iii) long-term feedbacks associated with ice sheets. ... biogeophysical/biogeochemical feedbacks (e.g., those associated with changes in methane, aerosols, ozone, or vegetation; Section 7.4.2.5) act both on time scales that are used to estimate the equilibrium climate sensitivity (ECS)."

  The implementation of these biogeophysical and biogeochemical processes varies among models. For instance, some models use static vegetation, some implement aerosol indirect effects, and others prescribe ozone and/or methane concentrations. Given this variability, we consider important to highlight this restriction in our manuscript.

- l. 82 paragraph: As mentioned before, please state which years are used for the regression of $\lambda_{ab}$;

  We used 150 years for the regression of $\lambda_{ab}$. This information has now been added to our manuscript on lines 73 and 85.

- l. 83 – 84: Is there a particular reason for subtracting the control state? I wonder, because a constant shouldn't affect the slope estimate. It wouldn't hurt the calculation, but I'm curious.

  There is no specific rationale for subtracting the control state, as a constant should not influence the slope estimate. To address

**this, we have revised the sentence in line 84 to: "We calculate forced climate feedbacks using linear Ordinary Least Squares (OLS) regression coefficients derived from 150 years of annual global averages of $R$ and $T$ from abrupt4xCO2 simulations."**

- l. 125: It is not immediately clear to me what was done here by "randomly permuting". Were the R and T time series randomly matched (e.g. R from model 1 realization 1 and T from model 2 realization 1), and were the feedback parameters subsequently computed from these randomly matched time series? Am I right in assuming that only complete time series were permuted, not individual values in the time series?

**We acknowledge that our initial description of the method lacked detailed explanation. To clarify, our approach did not involve randomly pairing the R and T time series to compute feedbacks from these pairs. Instead, the method estimates the likelihood of obtaining a correlation as high as, or higher than, the observed correlation between internal variability and forced climate feedbacks in climate models. Specifically, we randomly permuted the feedback datasets, disrupting the correspondence between models for internal variability and forced climate feedbacks (for example, by pairing the internal variability feedback from one model with the forced climate feedback from another). We then recalculated the correlation coefficient using this shuffled data. This procedure was repeated $10^5$ times, generating a null distribution of correlation coefficients that reflects the range of values expected if no real relationship exists. Finally, we compared the observed correlation to this null distribution to estimate how often a correlation of equal or greater magnitude could arise by chance, providing a p-value as a measure of statistical significance. We have added more details to this method in line 135 in the manuscript to clarify the procedure.**

- Fig. 2: I am not sure that Fig. 2 is really needed. To me as a reader, the only relevant information is the likelihood of obtaining the correlations by chance, which is mentioned in the text; the full distribution is not so interesting, and the differences between the blue, red, and black lines are anyway hard to grasp. While I take no issue with this figure, I believe that it could be removed without loss of information; however, I would like to see the likelihood to obtain the correlations for the net feedback parameter by chance in the text, I only found this information for LW and SW.

**We agree that the figure is not essential and have decided to remove it. We have added the likelihood of obtaining the correlations for the feedback parameters by chance in lines 144 and 155.**

- l. 150: Given that the first term is 0.43 and the last one is -0.72, does that mean that the internal SW feedback outperforms the internal LW feedback as a predictor for the forced LW feedback (by having a strong anticorrelation)? I find that interesting.

  **That is an interesting observation. However, it is important to clarify that the referred terms are not correlations, but rather the covariance divided by the product of standard deviations of the internal variability and forced climate net feedbacks. The actual correlation between internal variability and forced climate longwave feedbacks is 0.68, and the anticorrelation between shortwave internal variability and longwave forced climate feedback is -0.64. These values indicate that the internal variability shortwave feedback does not outperform the internal variability longwave feedback as a predictor of forced climate longwave feedback.**

- l. 174 – 175: So if the SW is the strongest contributor, that means that it comes down to clouds (unsurprisingly). Do you think the poor model representation of clouds is a problem for that?

  **Indeed, extending the period for estimating internal variability feedbacks improves the relationship between internal variability and forced climate shortwave feedbacks. Given that shortwave feedbacks are closely linked to clouds, this improvement indicates that clouds respond to natural variations in surface temperature similarly to how they respond to external radiative forcing in models. This consistent misrepresentation of clouds actually benefits our methodology as it allows us to use internal variability observations to constrain uncertainties in forced climate feedbacks. However, if models had more accurate cloud representations, the uncertainties in forced climate feedbacks might be reduced, potentially diminishing the applicability and need for this emergent constraint methodology.**

- l. 182: Models have no measurement uncertainty, but EBAF does. Is the uncertainty that arises from the satellite measurements (and also from the temperature data, but I assume that will be less important) taken into account? Would it affect the estimate of ECS or is it too small to make a difference? When combining the measurements from CERES and ERBE, is it problematic that the satellite changes, e.g., are there inconsistencies or steps?

  **We acknowledge that our current methodology does not explicitly incorporate satellite measurement uncertainties. In our analysis of internal variability feedbacks, the noise in the TOA fluxes time series arises from both natural variability and measurement errors. Consequently, when we regress TOA fluxes**

against surface temperature, the confidence intervals (CIs) inherently account for the total variability in the data, encompassing both natural fluctuations and measurement noise. Explicitly adding measurement uncertainties to the CI calculations could result in double-counting, thereby inflating the confidence intervals unnecessarily. This inflated observed internal variability feedback would, in turn, broaden the estimate of forced climate feedback after applying the emergent constraint. Consequently, using this broader estimate of forced climate feedback would lead to a wider range of ECS uncertainties.

Regarding the combination of satellite datasets, there are challenges involved, such as spatial biases in radiative fluxes, changes in the observing system used in the data assimilation process, and unrealistic variability in radiative fluxes due to the absence of volcanic aerosol effects. While these issues are indeed significant, they fall outside the scope of our study. Instead, we relied on the methodology implemented by Allan et al. (2014), who addressed these challenges and provided the merged dataset we used in our analysis.

- l. 187: The values are almost all well below 1 %. Doesn't that mean that less years might also be enough, if we think that, e.g., 5 % would be sufficient?

Indeed, fewer years might suffice; however, this is not the case for the most recent potential relationship, as illustrated in now Figures 2a and 2b. It would be interesting to verify whether the relationship holds for periods shorter than 51 years when including more years beyond 2014 if the historical simulations were extended. Nevertheless, we prefer not to include this observation to avoid speculation.

- Fig. 3 caption: Unclear what is meant by "n − 2014", what is n here? Should I read it as "n to 2014" or "n minus 2014"?

We clarify this now in the manuscript. The term "n − 2014" should be read as "starting year to 2014".

- l. 194 − 195: The suggested approach here is to wait for new satellite observations, but by then we will also have longer historical simulations. Can't we just run your analysis on the historical simulations again in 14 years, circumventing the whole problem of using the emergent relationship from one period with observations from another? It's still an interesting question to ask, but I don't see the practical necessity to use the "old" emergent relationship 14 years from now.

Yes, if we gather enough observations and extend the historical simulations to cover a 51-year period that includes the observational data, we could potentially apply the method without

relying on the "old" emergent relationship. We have now incorporated this consideration into line 212.

- l. 206 - 216: This seems to be in disagreement with the results of Fig. 4 (d). In Fig. 4 (d) you show that when taking at least 40 years, it doesn't matter which period one picks, $\lambda_{it}$ will always be the same. So $\lambda_{it}$ does not depend on the chosen period if the period is long enough. $\lambda_{ab}$ obviously doesn't depend on the chosen period either. So how can the relationship between $\lambda_{it}$ and $\lambda_{ab}$ depend on the chosen period (that's what I read from Fig. 4 a and b)? I have a hard time reconciling this. In addition, Gregory and Andrews 2016 (https://doi.org/10.1002/2016GL068406) show that historical feedback has varied quite a bit, although they use shorter than 40-year periods for their regression.

**We appreciate your observations and recognize that Figure 4 (Now Figure 3) may cause some confusion. In now Figure 3, we address two distinct questions with our analysis.**

**First, we test the hypothesis: "It is possible to use observations from one 51-year period with the model relationship between $\lambda_{it}$, from a different 51-year period, and $\lambda_{ab}$ to produce the emergent constraint." To validate this hypothesis, we used a proof by contrapositive approach, examining whether all model relationships using 51-year periods to estimate $\lambda_{it}$ would be statistically similar if the hypothesis were true.**

**In Figures 3a and 3b, we present the slopes and intercepts of all potential 51-year model relationships between 1850 to 2014 and compare them with the confidence intervals of their means. The results indicate that the 51-year relationships are indeed statistically different, providing evidence to reject the hypothesis. As the reviewer correctly notes, while $\lambda_{ab}$ remains unchanged, $\lambda_{it}$ varies for each 51-year period, indicating that those $\lambda_{it}$ from models are statistically different.**

**Second, we test the hypothesis: "It is possible to use the available 14 years of observations (2001-2014) with the model relationship between $\lambda_{it}$ (1958-2014) and $\lambda_{ab}$ to produce the emergent constraint." We again used a proof by contrapositive approach. If the hypothesis were true, then all possible 14-year $\lambda_{it}$ values within the period 1958-2014 would be statistically similar to the $\lambda_{it}$ of the full period 1958-2014. Using ERA5 reanalysis, we calculated all possible 14-year $\lambda_{it}$ values within the range 1958-2014 and compared them to that of the full period 1958-2014 (Figure 3c). The results provide evidence to reject the hypothesis. Additionally, we estimated the length of an observation period that would lead to a $\lambda_{it}$ statistically similar to that of the period 1958-2014, finding that 40 years are required (Figure 3c).**

In summary, the information presented in now Figure 3 addresses different questions and should be read with care. Figures 3a and 3b compare all modeled slopes and intercepts from the linear regressions between $\lambda_{ab}$ and 51-year $\lambda_{it}$ within the period 1850 and 2014 with their mean, while Figure 3c determines the observation window size needed to estimate a $\lambda_{it}$ statistically similar to that from the period 1958-2014 using data within the same period.

We would like to clarify that, upon reviewing the referee's comment, we realized our initial statement may have seemed categorical, implying that all 51-year modeled $\lambda_{it}$ periods are statistically different. In reality, there is some degree of similarity, as slopes and intercepts from adjacent time periods can be quite similar in certain cases. However, this pattern does not consistently apply across the entire analyzed period. Consequently, the extent of allowable discrepancy between the time periods used to calculate internal variability from observations and from models for generating an emergent constraint on forced climate feedbacks varies depending on the specific periods compared. We have updated the manuscript to reflect this clarification for improved understanding.

- Fig. 4 (a) and (b). How can the starting year be 1980 and higher for 51-year periods?

  **As noted by the reviewer, having a starting year of 1980 or later for 51-year periods is indeed inconsistent. We have identified and corrected an error in the computing code that led to this issue. The figures have been updated to reflect the correct starting year.**

- Does it surprise you that the relationship between $\lambda_{it}$ and $\lambda_{ab}$ varies strongly in time?

  **The substantial variation in the relationship between $\lambda_{it}$ and $\lambda_{ab}$ can be attributed to the previously mentioned error. Even after correcting for this, some variation persists, likely related to the specific internal variability present in each 51-year period. We chose not to include this in the manuscript as it remains speculative and requires further investigation beyond the scope of our study.**

- l. 250 − 252 and Fig. 5 (a): +/- 2 W/m$^2$ seems not negligible compared to interannual variability of global-mean TOA flux, which I would expect to vary by less than 10 W/m$^2$. How can it be that the correlation with CERES-ERBE is still so high (0.99)? It means that 98% of the variance of the ERA5 feedback parameter is explained by CERES-ERBE, so only

2 % is left for the error, which seems low given that the error gets up to +/- 2 W/m$^2$.

**We believe the referee is referring to the fact that 98% of the variance in the ERA5 TOA fluxes (rather than the feedback parameter) is explained by CERES-ERBE, leaving only 2% for the error. The correlation coefficient of TOA fluxes between ERA5 and CERES-ERBE measures the strength and direction of their linear relationship. A systematic offset between the two datasets does not significantly affect the correlation coefficient because it does not alter the way the datasets co-vary over time. Therefore, it's possible to have a high correlation coefficient (0.99) despite differences in their absolute values. Additionally, it is important to emphasize that a high R$^2$ value pertains to the variance in ERA5 TOA fluxes, not their exact values. Consequently, even with the absolute differences caused by the error margin, the relative variability and trends of the datasets remain aligned, resulting in a high correlation and coefficient of determination.**

- l. 277 – 279: I don't understand the method here. A probability density function of which quantity? What values are sampled from this distribution? I had expected one value for $\lambda_{it}$ from ERA5, obtained from regressing over the 40-year period, not a whole distribution. What am I missing? This seems like a central point of the paper and maybe deserves another sentence or two to clarify the method.

**We recognize that the initial description was difficult to read and included unnecessary details for deriving the emergent constraint. To clarify, we have simplified the explanation in the manuscript line 300.**

**In summary, our method uses a Monte Carlo simulation to estimate forced climate feedbacks. We begin by generating a predictor variable dataset from a truncated normal distribution, based on the 40 years of adjusted ERA5-CERES-ERBE observations and their confidence intervals. Applying a linear model ($\lambda_{ab} = m\lambda_{it} + b$), we calculate the confidence intervals for predicted forced climate feedbacks for each predictor value. To address prediction uncertainties, we sample from these confidence intervals, resulting in a new dataset of predicted forced climate feedback values. The emergent constraint is then characterized by the probability density function of this dataset.**

- Is there a reason for presenting the results from this analysis as small insets in Fig. 1? It seems like one of the main outcomes of this paper is hidden in a small inset. If showing it in Fig. 1, I would prefer the y-axes of the main plot and the inset to be aligned.

**After evaluating several alternatives, we determined that retaining the insets offers the clearest representation of feedback distri-**

**butions. To ensure consistency and improve visibility, we aligned the y-axes of the main plot and the inset.**

- l. 296 – 301: The list of limitations seems short. In addition to my questions about the pattern effect potentially limiting the results of this study, I think it may be beneficial to discuss further limitations. In particular, the emergent relationship is obtained from model simulations using models, hoping that this relationship would translate to the real world. However, most models that contribute to this relationship simulate $\lambda_{it}$ values way outside the observed range (see Fig. 1 f). Could this limit the results?

  **We have now expanded our discussion of limitations to include both the pattern effect and the assumption that emergent relationships from models apply to the real world. Concerning the issue of models simulating $\lambda_{it}$ values significantly outside the observed range, we previously noted that "uncertainties in the model emergent relationship, as illustrated in Figure 1f, reduce confidence in the emergent constraint". We have revised this statement for clarity to: "Uncertainties in the model emergent relationship within the adjusted ERA5-CERES-ERBE period, due to most models simulating $\lambda_{it}$ values significantly outside the observed range (Figure 1f), reduce confidence in the emergent constraint."**

**Minor comments:**

- l. 72 – 75: the half-sentence "incorporating a more extensive..." appears twice

  **The error has been corrected.**

- l. 161: The use of the word "assuming" makes sense here, but made me stumble, because it sounds like it's a prerequisite to run the hypothesis, when it's actually rather the null hypothesis; "testing for" or something similar would have been clearer to me.

  **We changed the word "assuming" to "where the null hypothesis posits" in line 180.**

**References**

Bloch-Johnson, Jonah et al. (2021). "Climate Sensitivity Increases Under Higher CO2 Levels Due to Feedback Temperature Dependence". In: *Geophysical Research Letters* 48.4, e2020GL089074. DOI: https://doi.org/10.1029/2020GL089074.

Forster, P. T. et al. (2021). "The Earth's Energy Budget, Climate Feedbacks, and Climate Sensitivity." In: *Climate Change 2021: The Physical Science Basis. Contribution of Working Group I to the Sixth Assessment Report of the Intergovernmental Panel on Climate Change.* Cambridge, United Kingdom and New York, NY, USA: Cambridge University Press. Chap. 7. DOI: 10.1017/9781009157896.009.

---

## Author Comment (AC2)

**Response to Referee Comments on "Constraining net long term climate feedback from satellite observed internal variability possible by mid 2030s"**

We appreciate the referees' constructive feedbacks. Below, we provide our detailed responses and the modifications made in response to their comments and suggestions.

- **Referee 2.**

The authors investigate the relationship between internal variability feedbacks and forced climate feedbacks across a range of CMIP6 models. They explore the feasibility of using this relationship, along with observed internal variability feedback estimates derived using CERES, to establish an emergent constraint on Equilibrium Climate Sensitivity (ECS). The authors find a robust relationship between internal variability and forced feedbacks, particularly for shortwave and longwave components, whereas the relationship seen for the net feedback is weaker. To address this, the authors explore how the relationship strengthens over longer time periods (50 years). To provide an estimated constraint on ECS, the authors combine satellite observations with a reanalysis dataset to provide an observed estimate of internal variability feedbacks. However, in order to provide a constraint based on observations only, continuous satellite observations until the mid-2030s would be necessary.

I found this paper enjoyable to read and I believe it would be a useful addition to the literature in this field. I have one major comment and a number of minor comments.

Major Comment:

The authors suggest that the relationship between internal variability feedback and forced feedback could be used as an emergent constraint on ECS. However, the utility of this relationship could be challenged were there to be a bias in modelled estimates of internal variability feedbacks compared to observations.

Armour et al. (2024) investigated the relationship between historical temperature trends and ECS, showing that this relationship was not suitable for use

as an emergent constraint due to a known systematic bias in modelled estimates of historical temperature trends.

They show that since coupled climate models do not simulate observed temperature patterns, modelled historical warming was systematically warmer compared to observations.

Would the results of Armour et al. (2024) impact the conclusions reached in this analysis? Would this suggest that there may be a systematic bias between observed and modelled internal variability feedbacks due to different SST patterns?

For example, if AOGCMs are biased in their simulation of SSTs patterns and feedbacks due to internal variability, then it is plausible that this biases the emergent constraint with long term feedbacks proposed here.

Either way, I would expect some discussion on the limitations and potential for biases in the results.

**The referee is asking whether the conclusions of Armour et al. (2024) —that coupled climate models, due to their inability to accurately simulate observed temperature patterns, render the relationship between modeled historical warming and ECS unsuitable as an emergent constraint— suggest that there may be a systematic bias between observed and modelled internal variability feedbacks due to different SST patterns.**

**As noted by Referee 1, these biases in warming patterns are more likely to lead to discrepancies between modeled and real-world forced climate feedbacks, rather than being directly tied to internal variability feedbacks. Such discrepancies may reflect differences in the relationship between internal variability and forced climate feedbacks in models versus the real world, which the emergent constraint methodology assumes do not exist. This is a caveat in our study and has now been explicitly highlighted in the abstract, discussion, and notes of caution in the revised manuscript.**

Minor Comments:

- Line 2-3 – Aren't the changes in top-of-atmosphere flux in response to surface temperature changes how we often define feedbacks in general (not just internal variability feedbacks). Could the definition of internal variability feedbacks and forced climate feedbacks be more explicitly defined?

    **You are correct in noting that changes in top-of-atmosphere flux in response to surface temperature changes are often how we define feedbacks in general. In our approach, the distinction between internal variability and forced climate feedbacks lies in the drivers of those surface temperature changes. For internal variability, changes in TOA fluxes result from natural variations in surface temperature due to internal variability. In contrast, for forced climate feedbacks, the changes in surface temperature**

**are driven by radiative external forcings. We have revised the sentece in line 2 to clarify this distinction. The phrase "top-of-the-atmosphere flux variations in response to surface temperature fluctuations" has been corrected to "top-of-the-atmosphere flux variations in response to natural surface temperature fluctuations"**

- Line 50 (and in the introduction in general) – I think it might be beneficial to formally define somewhere in the introduction what is meant by forced feedback and internal variability feedback.

  **The definition of forced climate feedbacks is now provided in line 21, while the definition of internal variability feedbacks is given in line 46. Details on their calculation begin in line 84.**

- Line 69 – Could the historical and amip experiments used be more clearly defined.

  **We have now defined the historical CMIP and AMIP experiments in the manuscript line 70.**

- Line 82 – "TOA flux anomalies" – Could the authors write "R" given they have shortened surface temperature anomalies to "T".

  **We have now replaced "TOA flux anomalies" with $R$.**

- Line 84 – Could the authors expand on how the historical members are detrended?

  **We removed the linear trend from each historical member individually to detrend the data. A more detailed description of this methodology has been added to the "Materials and Methods" section.**

- Line 85 – "various" – Could the authors be a bit more specific?

  **We have added a more detailed description of this methodology to the "Materials and Methods" section and the word "various has been removed.**

- Line 91 – "the transformed datasets" – It isn't completely clear what datasets are being referred to here.

  **We refer to the transformed temperature time series for each model realization. A more detailed description of this methodology has been added to the "Materials and Methods" section.**

- Line 81 Paragraph – I think in general, if this paragraph could be rewritten to be much more thorough with the details it would help. Further questions I am left with are... How is F calculated in order to calculate the lambda in the historical experiments? Is the idea that the internal variability feedbacks have no forcing effecting them? And if so, how is

this achieved given there is forcing over the historical period? Is this why the timeseries have been detrended? If the detrending has had a linear trend removed, given the historical temperature and flux timeseries' are often highly non-linear, is this definitely an appropriate method? Would that leave an imprint of the forcing still in the timeseries?

I think this paragraph was the main unclear bit in the paper for me.

I was also curious about whether this is a different method compared to that used in Uribe et al. 2022? (i.e. the use of GLS). Has that contributed to the slight change in the results?

**Thank you for your insightful questions. We address each point below:**

- **How is $F$ calculated to determine lambda in the historical experiments?: In our methodology, lambda is calculated as the regression slope between $R$ and $T$. Thus, $F$ is not explicitly needed for its calculation.**

- **Is the idea that the internal variability feedbacks have no forcing affecting them?: The concept of internal variability feedbacks pertains to the response of TOA fluxes to natural temperature variations driven by internal climate variability, rather than external forcing. Therefore, these feedbacks are considered independent of external radiative forcing.**

- **How is this achieved given there is forcing over the historical period?: We achieve this by removing the linear trend from the historical temperature and TOA fluxes time series to isolate internal variability. This detrending helps to eliminate the forcing effects in the relatively short time periods used in the study, allowing us to focus on short-term variability.**

- **If detrending has removed a linear trend, and given that historical temperature and flux time series are often non-linear, is this an appropriate method?: We understand the concern regarding non-linearity. Although radiative forcing is indeed non-linear over the extended period, the linear detrending approach is commonly used for short time scales. This method is supported by previous studies for similar periods (e.g., Uribe, Bender, and Mauritsen 2022; Chao and Andrew E. Dessler 2021; A. E. Dessler 2013; Lutsko et al. 2021; Mauritsen and Stevens 2015; Zhou et al. 2015).**

- **Would this leave an imprint of the forcing still in the time series?: In cases with large temporal datasets, there might be residual effects of forcing. However, in our study, we found that the leading EOF mode of CERES TOA variability (Figure 1a in this document) is highly correlated with**

the El Niño 3.4 temperatue anomalies (Figure 1b in this document), indicating that the variability in the data series is predominantly driven by internal variability rather than residual forcing effects.

[Figure]

Figure 1: a) Displays the dominant modes of variability in the CERES TOA flux dataset. b) Shows the principal component of the leading EOF mode of CERES TOA and the Niño region 3.4 temperature anomalies.

- **I was also curious about whether this is a different method compared to that used in Uribe et al. 2022? (i.e. the use of GLS). Has that contributed to the slight change in the results?: In our study, we analyze up to 5 realizations of historical ensemble members to capture a broader range of climate outcomes and obtain more robust estimates of internal variability feedbacks. This approach requires addressing autocorrelation in the temperature time series within each ensemble member, which we handle using GLS. Conversely, Uribe et al. (2022) used only a single ensemble member per model, so GLS was not needed in their analysis. Therefore, the differences in results between our study and Uribe et al. (2022) are primarily due to the use of multiple ensemble members rather than differences in the estimation methodology.**

- Line 125 – This isn't completely clear what has been done here. How were the datasets randomly permuted?

  **We acknowledge that our initial description of the method lacked detailed explanation, a point also noted by Referee 1. Here is our response: In summary, our method estimates the likelihood of obtaining a correlation as high as, or higher than, the observed correlation between internal variability and forced climate feedbacks in climate models. Specifically, we randomly permuted the feedback datasets, disrupting the correspondence between models for internal variability and forced climate feedbacks (for**

example, by pairing the internal variability feedback from one model with the forced climate feedback from another). We then recalculated the correlation coefficient using this shuffled data. This procedure was repeated $10^5$ times, generating a null distribution of correlation coefficients that reflects the range of values expected if no real relationship exists. Finally, we compared the observed correlation to this null distribution to estimate how often a correlation of equal or greater magnitude could arise by chance, providing a p-value as a measure of statistical significance. We have added more details to this method in line 135 in the manuscript to clarify the procedure. This methodology has been more clearly incorporated in now line 135.

- Figure 3 – All text in Figure 3 is very small and smaller details are rather hard to read. Could the authors increase the font size and perhaps consider re-arranging the subplot to help make it more readable.

  **We have rearranged now Figure 2 vertically to improve its readability.**

- Figures in general – Although the other figures are not as hard to read as Figure 3, some may benefit from larger text. Figure 5 is an example of this. Figure 5c also has the legend partially obscured by some of the lines in the plot.

  **We have resized now figure 4 for better readability.**

Very Minor Comments:

- Line 1 – ", crucial climate regulators," - I would remove this or restructure the sentence as it is a little unclear whether the authors are describing the act of observing climate feedbacks or the climate feedbacks themselves. Obviously it is the latter, but I kept reading it as crucial climate regulators is not feasible. – This is a very very minor comment, but I think it would help make the first line of the abstract more punchy.

  **We have reworded the sentence for better clarity.**

- Line 85 – "However" – This doesn't seem like quite the right word, "however" introduces a statement that contrasts or seems to contradict something. Would saying instead "Here, it is crucial...". Again, a very minor comment, but it stood out to me.

  **The sentence has been reworded.**

- Line 91 – "a OLS" –¿ "an OLS".

  **The spelling mistake has been corrected.**

**References**

Chao, Li-Wei and Andrew E. Dessler (2021). "An Assessment of Climate Feedbacks in Observations and Climate Models Using Different Energy Balance Frameworks". In: *Journal of Climate*, pp. 1–30. DOI: 10.1175/JCLI-D-21-0226.1.

Dessler, A. E. (2013). "Observations of Climate Feedbacks over 2000–10 and Comparisons to Climate Models". In: *Journal of Climate* 26.1, pp. 333–342. DOI: 10.1175/JCLI-D-11-00640.1.

Lutsko, Nicholas J. et al. (2021). "Emergent Constraints on Regional Cloud Feedbacks". In: *Geophysical Research Letters* 48.10, e2021GL092934. DOI: https://doi.org/10.1029/2021GL092934.

Mauritsen, Thorsten and Bjorn Stevens (2015). "Missing iris effect as a possible cause of muted hydrological change and high climate sensitivity in models". In: *Nature Geoscience* 8.5, pp. 346–351. DOI: 10.1038/ngeo2414.

Uribe, Alejandro, Frida A.-M. Bender, and Thorsten Mauritsen (2022). "Observed and CMIP6 Modeled Internal Variability Feedbacks and Their Relation to Forced Climate Feedbacks". In: *Geophysical Research Letters* 49.24. DOI: https://doi.org/10.1029/2022GL100075.

Zhou, Chen et al. (2015). "The relationship between interannual and long-term cloud feedbacks". In: *Geophysical Research Letters* 42.23, pp. 10, 463–10, 469. DOI: https://doi.org/10.1002/2015GL066698.

---

## Author Comment (AC3)

**Response to Community Comments on "Constraining net long term climate feedback from satellite observed internal variability possible by mid 2030s"**

We appreciate the community' constructive feedbacks. Below, we provide our detailed responses and the modifications made in response to his comments and suggestions.

- **Community comment.**

This is an interesting and useful study. However, I see two significant technical shortcomings in the authors' derivation of their emergent constraint based estimates of net long term climate feedback and of equilibrium climate sensitivity (ECS). The first of these two shortcomings biases the climate feedback estimate, weakening its central value by 32%, and together the two shortcomings bias the ECS estimate upwards by some 70%

1. The authors derive an emergent constraint on net long term climate feedback ($\lambda_{ab}$) with a median value of -1.56 $\mathrm{Wm^{-2}K^{-1}}$ from a linear regression fit between net internal variability feedback ($\lambda_{it}$) and $\lambda_{ab}$, as shown in Figure 1(f). I cannot see that the regression method used for this purpose is explicitly stated, but it appears to be standard ordinary least squares (OLS) regression of $\lambda_{ab}$ on $\lambda_{it}$. Such OLS regression, using data points and CERES-ERBE-ERA5 (observational) $\lambda_{it}$ of $-1.28$ $\mathrm{Wm^{-2}K^{-1}}$ digitized from Figure 1(f), yields a $-1.56$ $\mathrm{Wm^{-2}K^{-1}}$ central estimate for $\lambda_{ab}$, identical to that given in line 287.

   OLS regression assumes that the regressor variable is error free; if it is not then the regression slope will be biased towards zero ("regression dilution"). There is little uncertainty in the regressee variable, $\lambda_{ab}$, due to the high level of effective radiative forcing (ERF), and hence large changes in planetary net radiative balance (N) and surface temperature anomaly ($\Delta$T), involved in abrupt4xCO2 simulations by atmosphere-ocean global climate models (GCMs). However, there is significant uncertainty in the regressor variable, $\lambda_{it}$, as shown by the horizontal error bars in Figure 1(f). Hence OLS regression of $\lambda_{ab}$ on $\lambda_{it}$ is unsuitable and will give a slope estimate biased towards zero.

However, as there is little uncertainty in $\lambda_{ab}$, OLS regression will give an almost unbiased estimation if the regressor and regressee variables are switched, with $\lambda_{it}$ regressed on $\lambda_{ab}$.

Doing so gives a regression fit estimate of $\lambda_{it} = 0.187 + 0.637 \ \lambda_{ab}$, which on rearranging implies $\lambda_{ab} = 1.570 \ \lambda_{it} - 0.294$. The central estimate of $\lambda_{ab}$, based on the observed $\lambda_{it}$ estimate of $-1.28 \ \mathrm{Wm}^{-2}\mathrm{K}^{-1}$, is then $-2.30$ $\mathrm{Wm}^{-2}\mathrm{K}^{-1}$.

In summary, the authors should use $\lambda_{ab}$ rather than $\lambda_{it}$ as the regressor, in order to avoid significant bias in the estimated linear fit between them, and should adopt the resulting observationally-constrained $\lambda_{ab}$ estimate of $-2.30 \ \mathrm{Wm}^{-2}\mathrm{K}^{-1}$ in place of their $-1.56 \ \mathrm{Wm}^{-2}\mathrm{K}^{-1}$ estimate, which is seriously biased by regression dilution.

**To objectively assess whether the OLS regression used to characterize the emergent relationship between internal variability and forced climate net feedbacks is appropriate or significantly affected by regression dilution, we evaluate the performance of the OLS regression model by analyzing its residuals using several diagnostic methods. These include examining the mean residuals, the probability density function (PDF) (Figure 1 in this document), the Q-Q plot (Figure 2 in this document), and the residuals versus fitted values plot (Figure 3 in this document).**

**The mean residual (1.95e-16) and the probability density function (PDF) suggest that the model's residuals are centered around zero. Additionally, the Q-Q plot shows only minor deviations from normality, and the residuals versus fitted values plot reveals no distinct residuals pattern, indicating constant variance and minimal heteroscedasticity.**

**Furthermore, we repeated the regressions using Orthogonal Distance Regression (ODR), a method that accounts for errors in both the independent and dependent variables. The comparison of the regression coefficients from both methods (Table 1 in this document) shows minimal differences.**

**These findings indicate that, despite minor biases and deviations from normality, the model's fit remains robust, with minimal impact from regression dilution on the slope approaching zero. This evaluation alleviates concerns about regression dilution in calculating the emergent relationship, suggesting that our OLS regression results are likely reliable. We have now included this comparison in our revised manuscript.**

Figure 1: Probability density function (PDF) of residuals from the OLS regression between forced climate and internal variability net feedbacks. The distribution of residuals indicates a high likelihood close to zero, suggesting that the model's residuals are centered around zero, which supports the validity of the OLS regression model.

[Figure]

Figure 2: Q-Q Plot of Residuals from the OLS regression between forced climate and internal variability net feedback. The red line represents the theoretical quantiles, while the black markers show the sample quantiles. Minor deviations from the line indicate slight departures from normality.

[Figure]

Figure 3: Residuals against predicted values from the OLS regression between forced climate and internal variability net feedback. The lack of a clear pattern suggests that the residuals are randomly distributed around zero, indicating that the model's assumptions of linearity and homoscedasticity are reasonably met.

Table 1: Forced climate and internal variability feedbacks regression coefficients depending on the choice of regression method.

| | Longwave | | Shortwave | | Net | |
|---|---|---|---|---|---|---|
| | OLS | ODR | OLS | ODR | OLS | ODR |
| Slope | 0.57 | 0.57 | 0.74 | 0.74 | 0.57 | 0.62 |
| Intercept [$Wm^{-2}K^{-1}$] | -0.77 | -0.76 | -0.26 | -0.26 | -0.82 | -0.80 |

2. The standard estimate of a GCM's ECS corresponds to the $\Delta$T at which N = 0 when extending an OLS regression linear fit of annual mean N on $\Delta$T over 150 years of its abrupt4xCO2 simulation. That $\Delta$T mathematically equals minus the slope of the regression fit line, $-\lambda_{ab}$, divided into the value of N where the fit line intersects the $\Delta$T = 0 axis (F4x_reg150). It follows that the standard estimate is ECS = F4x_reg150 / $-\lambda_{ab}$.

For almost all GCMs, F4x_reg150 is significantly lower than the actual ERF from quadrupled CO2, as estimated from fixed SST simulations with a correction for land surface warming (F4x_SST-ts). The reason for this is simple. Net feedback is generally higher in the early part of 150 year abrupt4xCO2 simulations than in the much more numerous subsequent years, which have a dominant influence on linear estimation using OLS. As a result, in the earliest part of the N versus $\Delta$T plot the regression fit lies below the actual N values, most significantly when $\Delta$T = 0 at the start (taking F4x_SST-ts as the best estimate of the actual value of N at that point). As the standard estimate of ECS in GCMs (before scaling from 4x to 2x CO2) is F4x_reg150 / $-\lambda_{ab}$, ECS estimated as F4x_SST-ts / $-\lambda_{ab}$ will be biased upwards.

It follows that deriving ECS, as the authors do for their median ECS estimate of 2.5 K, by dividing an observationally-constrained GCM-based estimate of $-\lambda_{ab}$ into estimated F2x_SST-ts will significantly over-estimate ECS. This point is illustrated and more fully explained in sections 4.1 and S1 of Lewis (2023). Although using F2x_SST-ts rather than F2x_reg150 as the numerator when using estimated $\lambda_{ab}$ in the denominator is not uncommon when estimating ECS, doing so is unjustifiable: it is mathematically incorrect and causes significant overestimation of ECS.

On average, F4x_reg150 is 16% lower than F4x_SST-ts in the 17 CMIP6 models for which Smith et al (2020) were able to derive F4x_SST-ts (see their Table S1 ERFreg150 and ERF_ts values). Those 17 models have an average F4x_SST-ts of 8.41 Wm$^{-2}$. This should be converted to a value for a doubling of CO2, F2x_SST-ts, by dividing F4x_SST-ts by 2.10, per the formula in Meinshausen et al. (2020) that was adopted in IPCC AR6, rather than using the popular but inaccurate method of simply halving the 4x CO2 ERF. Doing so gives a F2x_SST-ts value of 4.01 Wm$^{-2}$ for the Smith et al (2020) CMIP6 mean. That is close to the 3.93 Wm$^{-2}$ value of F2x_SST-ts derived in AR6 and used in the manuscript. By contrast, F2x_reg150, the similarly converted value of F4x_reg150, is only 3.37 Wm$^{-2}$ for the 17 Smith et al (2020) models – almost identical to the 3.35 Wm$^{-2}$ average that I calculate for a larger set of 30 CMIP6 models.

The 2.10 ratio of quadrupled to doubled CO2 ERF is derived using detailed line-by-line radiation code, not simplified GCM radiation code, but the radiation code in CMIP6 models should be more accurate than that in earlier GCM generations. Moreover, I compute a similar (marginally higher) average 4x to 2x CO2 ERF ratio for the five GCMs analysed in Rugenstein et al (2020) with data from both abrupt4xCO2 and abrupt2xCO2

simulations, with the ratio being 2.10 for the only CMIP6 GCM included (CNRM-CM6). (That is based on estimating ERF by regression of N on $\Delta T$ over the first ten years after the abrupt CO2 increase, which provides a reasonable proxy for F4x_SST-ts in the Smith et al (2020) set of abrupt4xCO2 simulations.)

It follows that the authors should revise their ECS estimation formula to ECS = (F4x_reg150 / 2.10) /$\lambda_{ab}$, using the average regression-derived F4x_reg150 for the set of CMIP6 models used to constrain $\lambda_{ab}$. If that F2x_reg150 were the same as the 3.35 $Wm^{-2}$ that I calculated for 30 CMIP6 models, then the revised median ECS estimate, based on the corrected central $\lambda_{it}$ estimate of –2.30 $Wm^{-2}$K-1, would be 3.35 / 2.30 = 1.46 K. If the IPCC AR6 assessment of ECS is correct, then such a low ECS estimate may be considered unlikely to be accurate. If so, the correct, and important, conclusion to draw would then be that the relationship between $\lambda_{ab}$ and $\lambda_{it}$ in CMIP6 models does not provide a reliable emergent constraint on ECS.

**We recognize the concern regarding the use of estimated radiative forcing to calculate ECS. However, our primary objective is to explore how internal variability feedbacks can aid to constrain forced climate feedbacks. The ECS estimate presented in the manuscript serves as an example of what could be achieved with further observations of the Earth's radiation budget. We have emphasized that our emergent constraint results should be interpreted cautiously, as they offer indicative insights rather than definitive observational evidence.**

**In response to the community's feedback, we have included his suggested alternative approach in our revised manuscript. In addition to using the IPCC AR6 radiative forcing estimate, we now compute radiative forcing from the model set by taking the y-intercept of the regression between TOA flux anomalies and surface temperature anomalies, then dividing by 2.1. Using this alternative radiative forcing estimate and the constrained forced feedback, we provide the corresponding ECS estimate.**

The relationship of "true" ECS to that derived from regression over 150 years after a CO2 increase

Linear regression of N on $\Delta T$ over the first 150 years of an abrupt4xCO2 (or abrupt2xCO2) simulation, with ECS taken as the N = 0 intercept of the fit (ECSreg150) is the standard method for estimating the ECS of GCMs. Moreover, it is usual for observationally-constrained non-paleoclimate ECS studies to estimate that or another effective climate sensitivity measure. But, as noted in the Comment by Anonymous Referee #1, in GCMs

the actual (true) ECS, as estimated from ultra-long abrupt $CO_2$ increase forced simulations, generally exceeds ECSreg150. However, the 17% mean excess (for abrupt4x$CO_2$ simulations) stated in the paper they cited, Rugenstein et al (2020), includes the FAMOUS model, which appears to be near to runaway at quadrupled $CO_2$ – it warms almost four times as much as for doubled $CO_2$. On a forcing-adjusted basis (dividing abrupt4x$CO_2$ warming by 2.10), its 4x to 2x $CO_2$ ECS ratio is 1.86, while for all the other models with both simulations that ratio lies in the range 1.00 to 1.10.

The average excess of estimated actual ECS over ECSreg150 in the Rugenstein et al (2020) models excluding the outlier FAMOUS abrupt4x$CO_2$ simulation is 13.6%. The average ratio for the superset of those models included in Dunne et al (2020), ex FAMOUS, is almost identical.

Moreover, if true equilibrium ECS is to be estimated, it should logically be from ultra long abrupt2x$CO_2$ simulations, as the definition of ECS is for a doubling, not a quadrupling, of preindustrial carbon dioxide concentration. The average ratio, for the Rugenstein et al (2020) models, of estimated true ECS for 2x $CO_2$ to ECSreg150 derived by dividing abrupt4x$CO_2$ data by 2.10, is slightly lower at 1.11x (or 1.06x when including FAMOUS).

**We believe this community comment is referring to the main comment raised by Referee 1 and not specifically to our manuscript. Since we do not represent the referees' opinions directly in our work, we have chosen not to provide further commentary on this matter.**